# NePTune: A Neuro-Pythonic Framework for Tunable Compositional Reasoning on Vision-Language

**Danial Kamali**    **Parisa Kordjamshidi**
Michigan State University
{kamalida, kordjams}@msu.edu

## Abstract

Modern Vision-Language Models (VLMs) have achieved impressive performance in various tasks, yet they often struggle with compositional reasoning, the ability to decompose and recombine concepts to solve novel problems. While neuro-symbolic approaches offer a promising direction, they are typically constrained by crisp logical execution or predefined predicates, which limit flexibility. In this work, we introduce NePTune, a neuro-symbolic framework that overcomes these limitations through a hybrid execution model that integrates the perception capabilities of foundation vision models with the compositional expressiveness of symbolic reasoning. NePTune dynamically translates natural language queries into executable Python programs that blend imperative control flow with soft logic operators capable of reasoning over VLM-generated uncertainty. Operating in a training-free manner, NePTune, with a modular design, decouples perception from reasoning, yet its differentiable composition operations support fine-tuning. We evaluate NePTune on multiple visual reasoning benchmarks and various domains, utilizing adversarial tests, and demonstrate a significant improvement over base models, as well as its effective compositional generalization and adaptation capabilities in novel environments.

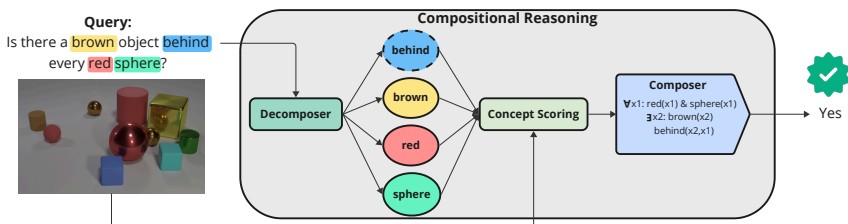

Figure 1: A natural language query (“*Is there a brown object behind every red sphere?*”) is *decomposed* into symbolic concepts, such as *red*, and *sphere*. These concepts are then *composed* to enable explicit reasoning over objects and their relations. This illustrates how complex queries can be mapped into structured logical forms.

## 1 Introduction

One key aspect of intelligence is the ability to compose known basic components to generalize and solve novel, complex problems. In vision-language reasoning, this capacity for compositional generalization is crucial. Humans easily decompose a complex query like *“Is there a brown object behind every red sphere?”* into its constituent concepts and compose them in order to reason about their relationships (Partee et al., 1984), as illustrated in Figure 1. However, modern vision-language models with transformer-based architectures often fail in dealing with novel compositions, revealing a gap between their pattern-matching skill and robust, human-like understanding (Zhu et al., 2022).

Recent studies have demonstrated the superiority of neuro-symbolic (NeSy) methods over monolithic neural architectures for reasoning beyond their observed data (Kamali et al., 2025; Yun et al., 2023). In recent years, methods such as VisProg (Gupta & Kembhavi, 2023) and ViperGPT (Surís et al., 2023) have leveraged Large Language Models (LLMs) to produce programs from language queries, breaking down complex questions into a sequence of executable steps. These programs are then run using pre-trained vision models for perceptual grounding. Although powerful, this paradigm has key shortcomings. First, these systems often rely on a sequence of crisp, intermediate decisions, creating brittle pipelines that are sensitive to perceptual errors. A single mistake in an early step (e.g., misidentifying an object) can cause the entire reasoning chain to fail. Second, these pipelines are inference-only and non-differentiable, which prevents them from adapting to new domains where pre-trained models may not perform well. On the other hand, methods such as LEFT (Hsu et al., 2024) and NeSyCoCo (Kamali et al., 2025) employ differentiable declarative symbolic reasoning, but their predicates are limited to the set of learned predicates from training on the target domain, which limits their zero-shot applicability. Table 1 summarizes this landscape and highlights where NePTune differs.

Table 1: Comparison of recent neuro-symbolic frameworks. We analyze the reasoning scope (local vs global), types of supported predicates, and predicate source. **VFM:** vision foundation models, e.g. XVLM; **VLM:** vision-language models, e.g. Qwen2VL.

| | | | Predicate Types | | | | Predicates | |
| --- | --- | --- | --- | --- | --- | --- | --- | --- |
| **Model** | **Reasoning** | **Supported Predicates** | **Class** | **Attr.** | **Rel.** | **Spatial** | **Pretrained** | **Trainable** |
| VisProg | Local | Predefined Modules | ✓ | ✓ | ✗ | ✓ | Modules | ✗ |
| ViperGPT | Local | Dynamic | ✓ | ✓ | ✗ | ✓ | VFMs | ✗ |
| LEFT | Global | Limited to Training Data | ✓ | ✓ | ✓ | ✓ | ✗ | ✓ |
| NeSyCoCo | Global | Limited to Similar Data as Training | ✓ | ✓ | ✓ | ✓ | ✗ | ✓ |
| NAVER | Global | Dynamic | ✓ | ✓ | ✓ | ✓ | VFMs | ✗ |
| **NePTune** | Hybrid | Dynamic | ✓ | ✓ | ✓ | ✓ | VLMs | ✓ |

To address the challenges of expressive compositional inference and domain adaptation, we propose NePTune, a neuro-symbolic visual reasoning framework. NePTune leverages an LLM to generate expressive Python programs where its execution does not solely rely on crisp decisions. Instead, it performs both soft compositional reasoning on the continuous scores provided by a VLM under uncertainty and imperative sequential reasoning. The result is a framework that decouples reasoning from the perception of atomic concepts, leading to remarkable generalization.

> **Contributions.**
>
> **1. A Hybrid Neuro-Symbolic Execution Model:** We propose a novel framework that seamlessly combines the imperative control flow of Python with soft compositional logic, enabling complex reasoning under uncertainty.
>
> **2. Domain Adaptable Framework:** We present a modular system that uses an LLM to generate programs on the fly, eliminating the need for predefined predicates and enabling zero-shot generalization as well as neuro-symbolic fine-tuning for domain adaptation.
>
> **3. Strong Compositional Generalization:** We demonstrate through extensive experiments that our approach significantly outperforms existing methods, particularly on challenging domain-shift benchmarks, showing highly robust results for vision-language reasoning.

## 2 RELATED WORK

NePTune is a framework that integrates multi-modal foundation models and logical reasoning to achieve compositional vision-language reasoning. Therefore, we focus on the three topics below.

### 2.1 COMPOSITIONAL VISION-LANGUAGE REASONING

Compositional reasoning is central to building reliable vision–language systems (Sinha et al., 2024; Hupkes et al., 2020), enabling models to represent and understand multi-step, relational structures (Ontañón et al., 2021). While VLMs achieve strong results across broad tasks (Xiao et al., 2023; Wang et al., 2024; Liu et al., 2024), diagnostic evaluations like CLEVR (Johnson et al.,

2017a) and ReaSCAN (Wu et al., 2021) expose persistent gaps in compositional generalization (Zhu et al., 2022) and brittleness under distribution shift (Yun et al., 2023; Li et al., 2025) in pure neural methods. To address this challenge, methods such as meta-learning (Xu et al., 2023), soft prompting (Xu et al., 2024), modified neural architectures Qiu et al. (2021); Kamali & Kordjamshidi (2023), and symbolic reasoning (Hsu et al., 2024; Kamali et al., 2025) have been explored. Among these methods, neuro-symbolic methods have been shown to be effective for compositional challenges (Kamali et al., 2025). Motivated by these findings, we adopt a neuro-symbolic approach that performs explicit compositional reasoning to achieve more robust and generalizable reasoning.

## 2.2 Neuro-Symbolic Approaches

Neuro-symbolic methods integrate neural perception with symbolic reasoning. Recent approaches utilize LLMs as semantic parsers to interpret linguistic queries (Gupta & Kembhavi, 2023; Faghihi et al., 2024). There are two common strategies for such methods. One line of work, including VisProg (Gupta & Kembhavi, 2023) and ViperGPT (Surís et al., 2023), generates imperative code for a sequence of local reasoning steps that rely on crisp decisions. While powerful, the sequential nature of these methods limits global reasoning under the uncertainty of these local decisions. The second strategy focuses on generating a declarative, symbolic representation of the query for robust symbolic reasoning. This line of work, as developed in NS-VQA (Yi et al., 2018), maps questions to functional programs executed over a structured scene representation in CLEVR-style synthetic worlds, using a fixed, dataset-specific inventory of attributes and relations and separately trained concept modules and a semantic parser. NS-CL (Mao et al., 2019) follows the same program-execution paradigm, but is trained end-to-end from question–answer supervision without annotated programs or attribute labels. LEFT (Hsu et al., 2024) extends this declarative approach with LLM-generated first-order logic programs and a differentiable executor, but still relies on domain-specific training data for concept learning and grounding, so its performance is tied to the concepts seen during training. NeSyCoCo (Kamali et al., 2025) improves lexical coverage with distributed predicate representations, but its generalization remains limited to concepts semantically close to those in the training distribution. NAVER (Cai et al., 2025), on the other hand, emphasizes orchestration through a finite-state controller that manages neural perception and utilizes ProbLog for global reasoning in referring expression grounding. However, our findings indicate that generating ProbLog queries is more error-prone for LLMs compared to Python code. Our aim in this work is to synthesize the advantages of both approaches. NePTune is a neuro-symbolic framework that performs both imperative and soft compositional reasoning over atomic predicates. Unlike works like NeSyCoCo and LEFT, the training is not necessary since the concept scores can be obtained zero-shot by harnessing the power of VLMs. However, with differentiable composition functions, it offers optional fine-tuning for domain adaptability by providing the capacity for both compositional inference as well as neuro-symbolic training.

## 2.3 Predicate Level Concept Understanding

A critical distinction between reasoning frameworks lies in how they derive and utilize concept-level understanding. General-purpose VLMs (Wang et al., 2024; Liu et al., 2024) learn concepts implicitly through end-to-end training, embedding this knowledge directly into the model's weights. In contrast, neuro-symbolic methods externalize this process. A program generating framework, such as VisProg, utilizes the concept grounding through a library of specialized, trained, hard-coded vision APIs, which return discrete labels or values. Similarly, ViperGPT utilizes foundation vision-language models, such as XVLM (Zeng et al., 2021), to obtain labels and discrete decisions. Another class of methods, including LEFT and NeSyCoCo, trains an MLP to generate continuous concept scores, representing the relations or attributes of an object in the visual regions. Our approach utilizes visual prompting, highlighting parts of the image via bounding boxes, to elicit referential concept scores from VLMs, which then serve as scores within our framework for reasoning under uncertainty.

## 3 Methodology

The NePTune framework performs visual reasoning via three core components, as shown in Figure 2. The process begins with the **LLM-based Program Generator**, which translates a natural

language query into both a Python program and a set of relevant object names. These names are passed to the **Perceptual Grounding** to detect all relevant candidate objects in the scene. Finally, the **Symbolic Executor** runs the generated program. During execution, it interacts with the Grounding Interface to obtain atomic concept scores and reason over them to get the final answer.

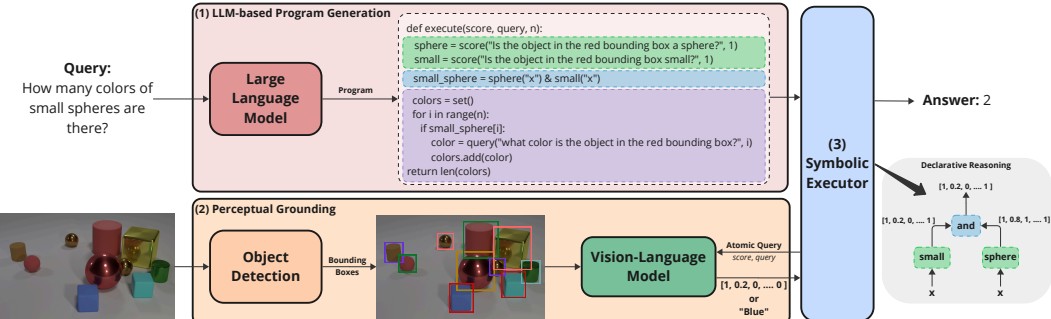

Figure 2: NePTune overview. Given an image and a query, the **(1) LLM-based Program Generation** converts the natural language query to a Pythonic program. Then **(2) Perceptual Grounding** extracts the object bounding boxes. The **(3) Symbolic Executor** then runs the Python code to reason over concepts extracted from the VLM using both soft composition and imperative logic to derive the final answer.

## 3.1 COMPONENT 1: LLM-BASED PROGRAM GENERATION

The first component of NePTune addresses semantic parsing. We leverage an LLM as a powerful few-shot parser to convert a natural language query into a formal, executable Python program. Given a query like "*Is there a big brown dog?*", the LLM is prompted to decompose it into a multi-step program that first identifies a set of candidate "dogs", and then reasons over the composition of atomic concepts such as "big", "dog", and "brown" to get the final answer. In the same LLM call, it also extracts object names for the region proposals. We chose Python as the target symbolic language for two key reasons. First, its Turing-complete nature, including rich control flow structures like loops and conditionals, is essential for expressing complex procedural reasoning. Second, the prevalence of open-source Python code makes LLMs particularly adept at generating it.

## 3.2 COMPONENT 2: PERCEPTUAL GROUNDING

This component connects the symbolic program to the visual world. It consists of two main parts: an object proposal module and a concept grounding module.

### 3.2.1 OBJECT PROPOSAL GENERATION

To ground objects in the scene to the concepts such as `blue` in the program, we must first identify all potentially relevant objects in the scene. We use the object names extracted by the LLM. For example, from "*Are there more cats to the left of the vase or dogs?*", the LLM extracts "dog", "vase", and "cat". We then feed these object names into Grounding DINO (Liu et al., 2023), a zero-shot object detection model that takes the image and the list of object names as input and outputs a set of bounding boxes for all matching objects. We note that while this textual query-based proposal is chosen for computational efficiency in real-world applications, our framework can operate with any reasonable region proposal method.

### 3.2.2 CONCEPT GROUNDING INTERFACE

Once a query is translated into a program, we must ground its atomic predicates to the visual content of the image, such as the objects in the scene. We handle this *perceptual grounding* using a VLM through two primary simplified interfaces **score** and **query**, where the image and bounding boxes are abstracted away with global variables for simplicity. The formal definitions of these grounding functions are as follows:

**score(query:str, num_objects:int)**: This function is the core of our reasoner. It takes a natural language question, an image, and related bounding boxes, and returns the concept probabilities for related object bounding boxes. This score is the VLM's normalized confidence for a "Yes" answer, computed from the logits of the "Yes" and "No" tokens. Formally, given an image $I$, set of $N$ detected objects, a visual prompt $v$, and atomic query $q_a$, the score $s$ is:

$$s = p(\text{"Yes"}|I, v, q_a) = \frac{e^{\text{logit}(\text{"Yes"})}}{e^{\text{logit}(\text{"Yes"})} + e^{\text{logit}(\text{"No"})}}$$

We treat these outputs as probability scores. To generate an object-centric concept score or answer via a VLM, we represent a symbolic predicate with a natural language question. For example, for the symbolic predicate `blue`, we use "*Is the object in the red bounding box blue?*". The bounding box $b$ provided to the functions will be drawn on target objects as a *visual prompt*, and it is controlled by the `num_objects` parameter, which specifies the predicate type:

- **num_objects=0**: For queries about the entire image (e.g., "*Is the photo taken indoors?*"). No extra visual prompt is used, and the output is a scalar probability.
- **num_objects=1**: For single-object queries, we mark the target object with a red bounding box around it as an extra visual prompt (e.g., "*Is the object in the red bounding box brown?*"). The output is a vector of size $N$.
- **num_objects=2**: For multi-object (relational) queries, we mark the main object with a red bounding box, and the secondary object with a green bounding box, as the extra visual prompt (e.g., "*Is the person in the red box talking to the person in the green box?*"). The output is an $N \times N$ matrix.

**query(query:str, object_id:int)**: This query function is used for tasks requiring open-ended answers. It takes a natural language question, an image, and an optional visual prompt, and returns a natural language string. Formally, given an image $I$, visual prompt $v$, and atomic query $q_a$, the generated answer $\hat{a}$ is :

$$\hat{a} = \arg\max_{a \in \mathcal{A}} p(a \mid I, v, q_a).$$

The answer $\hat{a}$ serves two purposes. First, it can provide the final output for questions that ask "*what*" or "*which*" (e.g., 'return query("*What color is the object in the red bounding box?*")'). Second, we use `query` to retrieve symbolic attributes as intermediate variables inside programs. For example, to count how many distinct shapes appear in a scene, the program can first ask for the shape of each object and then reason over the set of answers:

```
unique_shapes = set()
for i in object_ids:
  shape_i = query("What shape is object in red bounding box?", i)
  unique_shapes.add(shape_i)
num_unique_shapes = len(unique_shapes)
```

Here `query` returns short textual answers (typically a single token such as "cube" or "sphere"), which are then composed using standard Python control flow and data structures.

NePTune builds directly on top of VLMs that have been pretrained in an end-to-end fashion; our contribution is orthogonal to them and only changes how these backbones are used by exposing them as atomic concept scorers inside a symbolic executor instead of as direct prompting answerers.

### 3.3 COMPONENT 3: SYMBOLIC EXECUTOR

The final component of NePTune is the **Symbolic Executor**, which runs the LLM-generated program. A key innovation of our framework is its hybrid execution model, which integrates two distinct reasoning modes.

**1) Soft Compositional Reasoning:** To reason about visual concepts themselves, we employ a set of soft logical operations based on fuzzy logic principles. Instead of operating on binary true/false values, these operations work directly on the uncertainty scores obtained from the VLM. This is implemented through our custom data structures, which encapsulate the scores for a given predicate

and overload standard Python operators (`&` for AND, `|` for OR) to perform the corresponding logical operations. Details of these operations are shown in Table 2. For example, when the program executes `brown & dog`, our framework takes the element-wise minimum of the scores in the two corresponding tensors, implementing the fuzzy t-norm for conjunction.

Table 2: Soft compositional operators used in symbolic execution. Here, $\alpha$ represents an object-centric 1D score vector, $\beta$ relational kD ($k \geq 2$) score matrix, $\tau = 0.25$ is a temperature parameter, $\gamma = 0.25$ is a margin. We report default operator modes and training-time formulations.

| Syntax | Logical Form | Description | Differentiable Implementation |
|---|---|---|---|
| $\alpha_x.\texttt{exists()}$ | $\exists x \, \alpha_x$ | Existential quantification | $\max(\alpha_x)$ |
| $\alpha_x.\texttt{forall()}$ | $\forall x \, \alpha_x$ | Universal quantification | $\min(\alpha_x)$ |
| $\alpha_x \, \& \, \alpha_y$ | $\alpha_x \wedge \alpha_y$ | Logical conjunction | $\min(\alpha_x, \alpha_y)$ |
| $\alpha_x \, \& \, \beta_{xy}$ | $\alpha_x \wedge \beta_{xy}$ | Relational conjunction | $\min(\alpha_x, \beta_{xy})$ |
| $\alpha_x \, | \, \alpha_y$ | $\alpha_x \vee \alpha_y$ | Logical disjunction | $\max(\alpha_x, \alpha_y)$ |
| $\alpha_x.\texttt{implies}(\alpha_y)$ | $\alpha_x \rightarrow \alpha_y$ | Logical implication | $\max(1 - \alpha_x, \alpha_y)$ |
| $\texttt{not} \ \alpha_x$ | $\neg \alpha_x$ | Logical negation | $1 - \alpha_x$ |
| $\alpha_x.\texttt{iota}(\text{var})$ | $\iota(\text{var}, \alpha)$ | Best-match | $\frac{\alpha - \min \alpha}{\max \alpha - \min \alpha + \varepsilon}$ |
| $\alpha_x.\texttt{count()}$ | $\text{count}(\alpha_x)$ | Count | $\sum_x \alpha_x$ |
| $s_1 == s_2$ | $s_1 = s_2$ | Equality | $\sigma\left(\dfrac{2\gamma - |s_1 - s_2|}{2\gamma\tau}\right)$ |
| $s_1 > s_2$ | $s_1 > s_2$ | Inequality | $\sigma\left(\dfrac{s_1 - s_2 + \gamma}{\tau}\right)$ |

**2) Imperative Reasoning:** Our symbolic executor leverages a standard Python interpreter to handle the program's overall structure and control flow. This is possible since we have defined iteration and Boolean operations on the concept objects, allowing for complex, procedural logic, including conditionals (*if/else*), loops (*for*), and variable assignments, giving our framework the full expressive power of a general-purpose programming language. While some of these operations such as counting over string sets (as shown in Figure 2) can break the computation graph. This hybrid design enables the system to reason fluidly under uncertainty while fully leveraging the expressive power of a general-purpose programming language.

## 4 EXPERIMENTS

We structure our experiments around four research questions. **RQ1:** How does NePTune perform on zero-shot compositional reasoning on synthetic data? (Experiment 1), **RQ2:** How does NePTune perform on complex human-generated questions? (Experiment 2), **RQ3:** How does NePTune perform in grounding referring expressions in natural images? (Experiment 3), and **RQ4:** How does NePTune perform under domain shift and adapt to an unseen environment? (Experiment 4).

### 4.1 EXPERIMENTAL SETTING

We evaluate NePTune on four benchmark groups to cover diverse reasoning aspects. First, to assess zero-shot compositional reasoning on *synthetic data*, we employ CLEVR (Johnson et al., 2017a) and its extensions (Hsu et al., 2024) (Ref, Puzzles, and RPM), comparing NePTune against monolithic VLMs (Qwen2VL (Wang et al., 2024), InternVL2.5 (Chen et al., 2025), Ovis1.6 (Lu et al., 2024)), and symbolic methods such as LEFT, NeSyCoCo, and ViperGPT. Second, to evaluate performance on *complex human-generated* questions, we use the CLEVR-Humans (Johnson et al., 2017b) dataset and compare against similar baseline as synthetic CLEVR. We also include the trained neural model MDETR (Kamath et al., 2021) to compare NePTune against neural methods under linguistic diversity. Going beyond VQA, we evaluate NePTune on the task of *grounding of referring expressions in natural images*, we use RefCOCO-Adversarial (Akula et al., 2020) (Ref-Adv), with baselines including direct VLM grounding, specialized systems such as Grounding DINO (Liu et al., 2023) and Florence2 (Xiao et al., 2023), ViperGPT, and NAVER. Finally, to measure performance under domain shift and the ability to adapt to unseen environments, we use the photo-realistic gaming dataset Ref-GTA (Tanaka et al., 2019), comparing NePTune against both zero-shot and fine-tuned backbone VLMs, specialized grounding models, ViperGPT, and NAVER. Table 9 provides a high-level summary of NePTune's gains over these backbones across benchmarks, and Appendix D provides the details on all the datasets and preprocessing. The code is publicly available at `https://github.com/HLR/NePTune`.

## 4.2 EXPERIMENT 1: CORE COMPOSITIONAL REASONING

Table 3: Accuracy comparison across CLEVR question categories. Results are shown by reasoning type and model paradigm, including zero-shot, neural, and neuro-symbolic approaches.

|  | **InternVL2.5** | **NePTune** | **ViperGPT** | **NeSyCoCo** | **LEFT** |
|---|---|---|---|---|---|
| **Training Category** | Zero-Shot Neural | Zero-Shot NeSy | Zero-Shot NeSy | Trained NeSy | Trained NeSy |
| Final Accuracy | 90.25 | **92.65** (↑ 2.40) | 36.05 | 99.68 | 99.50 |
| Exist | 87.10 | **93.19** (↑ 6.09) | 48.75 | 99.28 | 98.92 |
| Query Attribute | **98.26** | 96.81 (↓ 1.45) | 29.42 | 100.00 | 99.86 |
| Compare Attribute | **98.61** | 91.94 (↓ 6.67) | 53.06 | 99.44 | 99.72 |
| Count | 74.60 | **87.10** (↑ 12.50) | 21.37 | 99.79 | 98.99 |
| Compare Number | 90.86 | **92.57** (↑ 1.71) | 48.57 | 100.00 | 100.00 |

To evaluate NePTune's compositional reasoning capabilities (**RQ1**), we evaluate its performance on the **CLEVR** benchmark. CLEVR is a standard benchmark for compositional reasoning that features synthetic 3D-rendered images and questions testing compositional visual reasoning. Details of the question categories are provided in Appendix D. The results in Table 3 show that, within the family of neuro-symbolic systems, NePTune establishes itself as the strongest zero-shot method, achieving 92.65% accuracy compared to only 36.05% for ViperGPT. While trained approaches such as NeSyCoCo and LEFT nearly saturate CLEVR (99%), NePTune demonstrates that competitive compositional reasoning can be achieved without dataset-specific supervision. Compared to its backbone VLM, InternVL2.5, NePTune still yields improvements raising overall accuracy from 90.25% to 92.65% (↑ 2.40%). The largest gains appear in quantitative categories where explicit compositional structure is most useful: *Count* rises from 74.60% to 87.10% (↑ 12.50%), and *Compare Number* increases from 90.86% to 92.57% (↑ 1.71%). We also see a notable improvement on *Exist* from 87.10% to 93.19% (↑ 6.09%), consistent with the executor reducing spurious correlation when filtering by attributes and relations. In contrast, attribute-heavy categories regress: *Query Attribute* drops to 96.81% (↓ 1.45%) points and *Compare Attribute* drops to 91.94% (↓ 6.67%) points. A closer look at concept-level accuracy revealed that analogical concepts such as *same color* or *same shape* remain among the most challenging to capture in this benchmark. More discussion of atomic concept evaluation is provided in Section 5.

Table 4: Accuracy on CLEVR extension tasks. Methods marked with [†] use ground-truth programs. Improvements from the backbone VLM (InternVL2.5-8B) are marked with the arrow sign (↑).

|  | **Method** | **Ref(%)** | **Puzzles(%)** | **RPM(%)** |
|---|---|---|---|---|
| Trained | NeSyCoCo[†] | **100.00** | **95.00** | **100.00** |
|  | NeSyCoCo | 94.00 | 94.00 | 74.00 |
|  | LEFT | 94.00 | 85.00 | 87.00 |
| Zero-shot | Qwen2VL-7B | 21.00 | 43.00 | **53.00** |
|  | InternVL2.5-8B | **27.00** | **52.00** | 47.00 |
|  | Ovis1.6-9B | 4.00 | 47.00 | 49.00 |
|  | ViperGPT | 8.00 | 34.00 | 4.00 |
|  | NePTune[†] | **99.00** (↑ 72) | **85.00** (↑ 33) | **97.00** (↑ 50) |
|  | NePTune | 91.00 (↑ 64) | 81.00 (↑ 29) | 80.00 (↑ 33) |

In addition to the original CLEVR benchmark, we evaluate our method on various compositional challenges based on the CLEVR environment introduced in LEFT, including referring expressions (**CLEVR-Ref**), visual puzzles (**CLEVR-Puzzles**), and Raven's Progressive Matrices (**CLEVR-RPM**). As shown in Table 4, NePTune shows the highest performance among the zero-shot methods and outperforms the direct prompting VLMs. In addition, compared to NeSyCoCo and LEFT with trained concepts, we show competitive performance.

## 4.3 EXPERIMENT 2: COMPLEX HUMAN QUERIES

To evaluate NePTune on more complex and diverse human-generated language (**RQ2**), we utilize the **CLEVR-Humans** (Johnson et al., 2017b) benchmark to compare NePTune against other neuro-symbolic and pure neural models. As shown in Table 5,

NePTune significantly outperforms prior declarative neuro-symbolic methods such as LEFT and NeSyCoCo by a large margin of ↑ 31.55% and performs better than the MDETR trained neural method. It also surpasses the imperative ViperGPT. Furthermore, it improves upon its powerful monolithic backbone (InternVL2.5-8B) by ↑ 1.72%, illustrating its effectiveness on complex, human-generated questions. In addition, to demonstrate the added value of NePTune's expressivity, we conducted an experiment with *NeSyCoCo + VLM*, which leverages the same InternVL2.5 as the concept-sco ring module. While this experiment yields a ↑ 12.36% improvement over NeSyCoCo, it leaves a large gap behind NePTune. This gap indicates the advantage of our expressivity gained by our method.

Table 5: Accuracy on the CLEVR-Humans (CH).

| | Method | CH (%) |
|---|---|---|
| Trained | LEFT | 56.69 |
| | NeSyCoCo | 56.12 |
| | MDETR | **81.73** |
| Zero-shot | Qwen2VL-7B | 84.12 |
| | InternVL2.5-8B | **85.95** |
| | Ovis1.6-9B | 79.96 |
| | ViperGPT | 31.05 |
| | NeSyCoCo + VLM | 68.48 |
| | NePTune | **87.67** |

## 4.4 EXPERIMENT 3: REAL-WORLD IMAGES

While various versions of CLEVR used in the aforementioned experiments are strong testbeds for compositional reasoning, after all, these are toy environments with limited diversity. To demonstrate NePTune's ability to operate in realistic environments (**RQ3**), we evaluate it on Referring Expression Grounding (REG) of real-world images using the **RefCOCO-Adversarial** (Akula et al., 2020) (Ref-Adv) benchmark. On Ref-Adv, NePTune is the strongest zero-shot reasoner among symbolic baselines such as ViperGPT and NAVER. To ensure a fair comparison with NAVER, we follow its setup and use the same backbones (Grounding DINO + XVLM + InternVL2-8B), denoted as NePTune‡. Under this setting, NePTune‡ reaches 63.71 vs. 36.45 for NAVER† (execution only), and with a simple verification (similar to NAVER) attains 75.54 vs. 65.13 for NAVER. Compared to pure neural methods, NePTune surpasses all the specialized grounding methods, such as Grounding DINO and Florence2 (Xiao et al., 2023). Additionally, compared to their VLM backbones, both NePTune and NePTune‡ with verification surpass their respective backbones. Details on verification are available in Appendix E.

Table 6: Accuracy on Ref-Adv dataset.

| Method | Ref-Adv (%) |
|---|---|
| Grounding DINO-B | 60.85 |
| Florence2-L | 71.73 |
| Ovis1.6-9B | 58.47 |
| InternVL2-8B | 72.92 |
| InternVL2.5-8B | 76.13 |
| ViperGPT | 60.66 |
| NAVER† | 36.45 |
| NAVER | 65.13 |
| NePTune‡ | 63.71 |
| + Verification | 75.54 |
| NePTune | 71.57 |
| + Verification | **78.08** |
| NePTune (1B) | 60.69 |
| + Fine-tuning | 68.06 ± 0.56 |
| + Verification | 74.59 ± 0.12 |

## 4.5 EXPERIMENT 4: GENERALIZATION AND ADAPTATION

When using popular benchmarks, as we did in Experiments 1-3, the possible data contamination makes zero-shot performance less reliable. To answer **RQ4**, we test NePTune's ability to generalize and adapt to novel environments using a less popular photo-realistic gaming environment that is **Ref-GTA** (Tanaka et al., 2019) benchmark. Ref-GTA is a challenging REG benchmark with images from a game simulation, which creates a significant domain shift from the natural images on which most VLMs are pre-trained. Based on the reports of Ovis1.6 and InternVL2.5, this source is not included in their training data. As shown in Table 7, this domain shift causes a significant performance drop for most methods. The powerful pure neural InternVL2.5-8B, for example, fails catastrophically, with its accuracy dropping to 6.95%. In contrast, NePTune, when paired with the same VLM, achieves a remarkable 69.69% accuracy. This demonstrates

Table 7: Accuracy on Ref-GTA benchmark.

| Method | Ref-GTA(%) |
|---|---|
| GroundingDINO-B | 27.90 |
| Florence2-L | 58.65 |
| Ovis1.6-9B | 2.78 |
| InternVL2.5-8B | 6.95 |
| InternVL2.5-1B | 1.64 |
| + Fine-tuning | 32.61 ± 0.35 |
| ViperGPT | 1.40 |
| NAVER† | 54.84 |
| NAVER | 58.73 |
| NePTune‡ | 62.73 |
| NePTune | **69.69** |
| NePTune (1B) | 34.92 |
| + Fine-tuning | 69.90 ± 1.16 |

that by utilizing our global symbolic reasoner and atomic-level concept understanding, our framework achieves robustness and compositional generalization that monolithic models lack. Furthermore, while NePTune demonstrates strong zero-shot robustness, its soft operations also enable fine-tuning. As shown in Table 7, by fine-tuning a smaller VLM (InternVL2.5-1B) using our neuro-symbolic differentiable computations with only 1000 samples, we can further improve its performance and obtain 69% accuracy. While fine-tuning VLM using original neural training, it only reaches 32% accuracy. These findings are promising for future

research and using neuro-symbolic reasoning as a source of supervision for larger-scale VLM training. More details on fine-tuning are presented in Appendix F.

## 5 DISCUSSION

**VLM for Concept Understanding.** A central idea of NePTune is to employ VLMs as underlying perception modules and concept grounders. The basic assumption here is that while VLMs are prone to fail at reasoning over complex compositions, they should perform better in perceiving basic concepts. However, basic perception questions require isolating parts of the visual input, a process known as visual prompting, which has been shown to be challenging in complex questions (Cai et al., 2024). In this section, we aim to evaluate the hypothesis that modern VLMs guided by visual prompts are effective as concept grounders for our purpose. We utilize ground-truth scene graphs from the **CLEVR** (Johnson et al., 2017a) and **Visual Genome** (Krishna et al., 2016) datasets to automatically generate a set of atomic templated questions. For CLEVR, we create a benchmark using scene graphs of 200 sampled scenes, and for Visual Genome, we randomly sample 1000 questions from 200 scene graphs. We generate simple Yes or No questions about their class, attributes, and relations, such as, "*Is the object in the bounding box a bird?*". We then use our `score` function that employs the underlying VLM to answer these atomic questions.

Table 8: Backbone VLMs performance on CLEVR and VG scene graphs (Micro F1-Score x 100)

| Category | InternVL2.5-8B | | Ovis1.6-9B | | Qwen2VL-7B | |
|---|---|---|---|---|---|---|
| | **CLEVR** | **VG** | **CLEVR** | **VG** | **CLEVR** | **VG** |
| Attribute | **97.25** | **89.37** | 95.87 | 86.75 | 89.27 | 83.45 |
| Class/Object | 90.00 | 81.76 | **91.98** | 80.42 | 82.21 | **83.28** |
| Relation | **90.94** | 82.35 | 87.91 | 80.00 | 80.04 | **83.02** |
| Spatial | **89.82** | 93.42 | 86.83 | 93.39 | 89.53 | **94.41** |
| **Overall** | **94.54** | 90.41 | 90.35 | 88.19 | 86.59 | **90.77** |

The results are shown in Table 8. Our analysis reveals that VLMs are particularly strong at identifying objects, classes, properties, and spatial relations. However, the performance is weaker on more complex relational concepts, especially for analogical comparisons such as *same size*. This problem is more severe in the Qwen2VL model. Although visual prompting can be problematic for general VQA (Cai et al., 2024), our results show this is not as challenging in our setting for grounding atomic queries. These results demonstrate that VLMs perform significantly better in answering atomic queries compared to complex multi-step queries, as reported in Tables 4 and 5. For example, there is a performance gap of up to 67% when comparing CLEVR-Ref (27%) compared to CLEVR average atomic evaluation (94%) using InternVL2.5. Detailed concept-level results of the experiment are presented in the Appendix B. Also, the results in Table 9 show the performance of different VLMs in NePTune, where they show an average improvement of up to ↑ 32%.

Table 9: Performance of different VLM baselines with direct prompting compared to NePTune across benchmarks. Δ values indicate gains or losses relative to the raw VLMs.

| Model | CLEVR (%) | Ref-Adv (%) | Ref-GTA (%) | Ref (%) | Puzzles (%) | RPM (%) | Avg. (%) |
|---|---|---|---|---|---|---|---|
| InternVL2.5-8B | 90.25 | 76.13 | 6.95 | 27.00 | 52.00 | 47.00 | 49.89 |
| + NePTune | 92.65 ↑ 2.40 | 78.08 ↑ 1.95 | 69.69 ↑ 62.74 | 91.00 ↑ 64.00 | 81.00 ↑ 29.00 | 80.00 ↑ 33.00 | 82.07 (↑ 32.18) |
| Ovis1.6-9B | 85.00 | 58.47 | 2.78 | 4.00 | 47.00 | 49.00 | 41.04 |
| + NePTune | 88.80 ↑ 3.80 | 63.25 ↑ 4.78 | 56.32 ↑ 53.54 | 85.00 ↑ 81.00 | 69.00 ↑ 22.00 | 80.00 ↑ 31.00 | 73.73 (↑ 32.69) |
| Qwen2VL-7B | 93.55 | 81.07 | 1.75 | 21.00 | 43.00 | 53.00 | 48.90 |
| + NePTune | 82.55 ↓ 11.00 | 80.93 ↓ 0.14 | 68.87 ↑ 67.12 | 81.00 ↑ 60.00 | 65.00 ↑ 22.00 | 71.00 ↑ 18.00 | 74.89 (↑ 25.99) |

**Choice of Symbolic Program.** We analyze program execution success rates (no syntax or runtime errors) on 500 Ref-Adv and CLEVR-Humans using the GPT-4o (OpenAI-Team, 2024) LLM. As shown in Table 10, frameworks that rely on specialized, non-Pythonic syntax, such as LEFT, NeSyCoCo, and NAVER (which uses

ProbLog), frequently suffer from generation failures. We found that the LLM often struggles to correctly translate natural language into their specific, strict formalisms. In contrast, methods that utilize Python, such as ViperGPT and NePTune, demonstrate significantly higher rates of successful program execution. However, our analysis of ViperGPT's failures reveals that its purely imperative approach, which often relies on selecting objects by a fixed index, is a primary source of error. The NePTune hybrid reasoner proves to be the most robust. Our analysis shows that the main source of error for NePTune shifts from low-level syntax errors to higher-level logical errors, where the LLM fails to compose the correct sequence of predicates. Qualitative examples are shown in Appendix A.

Table 10: Execution success rate (%) of different neuro-symbolic methods on Ref-Adv and CLEVR-Humans.

| Method | Ref-Adv | CLEVR |
|---|---|---|
| NeSyCoCo | N/A | 70.33±3.47 |
| LEFT | N/A | 64.33±3.94 |
| ViperGPT | 48.67±1.70 | 95.42±0.30 |
| NAVER | 23.02±4.71 | N/A |
| NePTune | **98.66**±0.82 | **97.24**±0.85 |

To quantify the correctness of NePTune's programs, we conducted a human evaluation of 600 generated programs across the Ref-Adv, Ref-GTA, and CLEVR-Humans datasets (see Appendix H). The results indicate that syntax errors are rare in the generated Python programs. Instead, most failures arise from higher-level logical composition mismatches rather than low-level execution issues. On real-world referring expression datasets, the dominant errors occur when atomic prompts fail to preserve plural or group-level concepts from the original natural language query. On the CLEVR-Humans VQA dataset, the few observed errors primarily stem from incorrect application of normalization operators such as `iota`. Overall, evaluators judged at least 87% of the programs to be logically sound, indicating that NePTune not only mitigates brittle execution failures but also that its first-order logic interface facilitates the generation of logically sound reasoning programs.

**Latency and Computational Efficiency.** The primary computational cost of NePTune arises from the calls to the VLM for concept grounding, which scales at $O(RN^2)$ for a scene with $N$ candidate objects and $R$ binary relations. To mitigate this overhead, we employed batched prefix caching with the vLLM (Kwon et al., 2023) tool. Since image tokens and most textual prefixes are shared across atomic queries, this caching mechanism eliminates redundant computations. As detailed in Appendix G, this optimization reduces the average execution time on the Ref-Adv dataset. While a long tail of highly complex, multi-object queries exists that increases the overall average, the majority of queries are resolved in under 5s for the 8B model and under 2s for the 1B model. This demonstrates that NePTune introduces a controlled, acceptable overhead compared to a single VLM forward pass, offering a favorable trade-off for gains in compositional robustness.

## 6    CONCLUSION AND FUTURE WORK

In this work, we introduce NePTune, a novel neuro-symbolic framework for compositional vision-language reasoning. Our approach leverages a novel hybrid reasoner that combines the imperative control flow of Python with a declarative, probabilistic soft logic that operates directly on the uncertain outputs of a VLM. This design enables our framework to be both highly expressive and robust in the face of perceptual uncertainty. Our extensive experiments demonstrate the effectiveness of this approach. NePTune demonstrates clear improvement over strong baselines, as well as the underlying VLMs, on complex compositional reasoning benchmarks, in visual question answering, and referring expression, exhibiting remarkable generalization capabilities. While the performance of NePTune is dependent on the quality of its underlying components, such as the LLM for program generation and the VLM for concept grounding, our work demonstrates a flexible and powerful paradigm for building more robust and generalizable AI systems, leveraging these models to elevate their capabilities to a comparatively higher level of performance. Future work could explore methods for more efficient grounding by incorporating external knowledge graphs to resolve complex, non-visual commonsense relations. Additionally, integrating NePTune's hybrid symbolic reasoning into broader declarative neuro-symbolic frameworks (Faghihi et al., 2021) or sophisticated orchestrations (Cai et al., 2025) could enable the enforcement of explicit logical constraints during fine-tuning (Guo et al., 2020) and further elevate the robustness of visual reasoning.

ACKNOWLEDGMENTS

This project is supported by the Office of Naval Research (ONR) grant N00014-23-1-2417. Any opinions, findings, and conclusions or recommendations expressed in this material are those of the authors and do not necessarily reflect the views of the Office of Naval Research.

ETHICS STATEMENT

This work complies with the ICLR Code of Ethics. NePTune builds on publicly available datasets and open-source VLMs, without collecting new human-subject data. Large Language Models were used solely for grammar polishing and minor text improvements.

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

## A  QUALITATIVE ANALYSIS

### A.1  REFERRING EXPRESSIONS GROUNDING

Examples of NePTune on Ref-GTA are shown in Figure 3. **Expressions 1 and 2** show the superiority of compositional reasoning compared to the VLM and object detection backbones. They demonstrate how the VLM fails to correctly locate the object in the new environments, while atomic concepts are correctly scored by the VLM.

### A.2  VISUAL QUESTIONS ANSWERING

Examples of NePTune on CLEVR-Humans are shown in Figure 4. **Example 1** illustrates a case where reasoning over meta-concepts such as distinct shapes is required. NeSyCoCo is unable to resolve the query due to this limitation to declarative, while ViperGPT struggles to identify the correct anchor object because it does not perform global reasoning across the object set. NePTune, by contrast, is able to provide the correct interpretation. **Example 2** presents a more complex scenario where reflection reasoning and calibration are critical. Both the sphere and the cylinder are reflected in the cyan cube, with the cylinder's reflection being more prominent and thus the more likely answer. NePTune does not fully capture this subtle distinction and produces an incorrect response. The competing baselines, however, fail for different and more fundamental reasons: VLM score calibration issues lead to incorrect selection, and ViperGPT relies on bounding box size comparisons that ignore perspective, which causes it to treat equally sized cubes as different. In this case, the code selects the shiny brown cube in the foreground and concludes that there is no reflection. Although NePTune also errs, this example highlights the inherent difficulty of fine-grained reflection reasoning and the challenges posed by subtle visual cues.

### A.3  FAILURE EXAMPLE.

Example shown in Figure 5 illustrates an error caused by an incorrect LLM-generated program in NePTune. Two issues occur simultaneously:

1. **Incorrect predicate arity:** The program invokes the function `is_baby_giraffe` with `num_object=2` instead of `1`. This leads to a dimensional mismatch when the resulting score tensor is composed with variable `x2`, causing execution failure.

2. **Incorrect bounding box reference:** The program refers to the bounding box with the color *green* instead of *red*, which is the intended single-object reference.

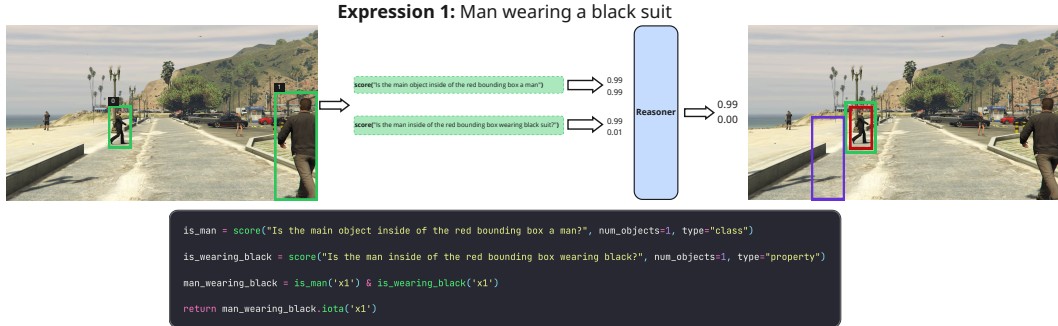

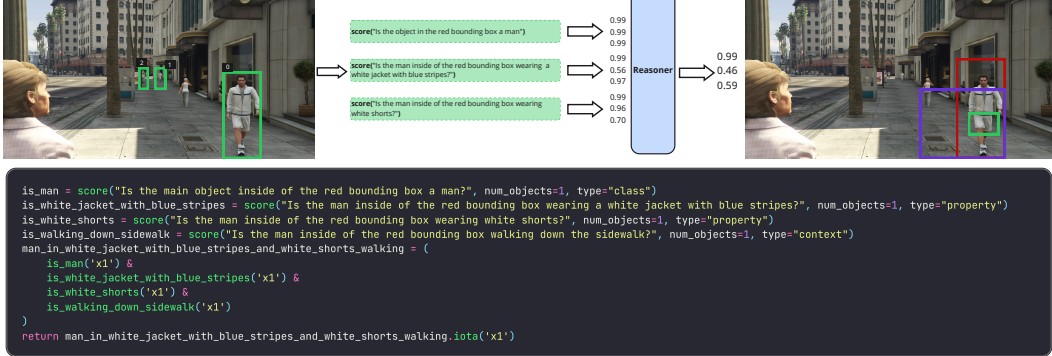

Figure 3: Qualitative examples of NePTune on the RefGTA dataset. Green boxes indicate objects detected by Grounding DINO, blue boxes show objects selected by the VLM (InternVL2.5-8B), and red boxes highlight the final selections made by NePTune.

While errors such as incorrect bounding box colors can be alleviated to a certain extent with regex-based matching, the dimensional inconsistency in predicate scoring is unrecoverable without regeneration.

## B  VLM CONCEPT ANALYSIS

### B.1  CONCEPT-LEVEL PERFORMANCE

An analysis of the atomic concept grounding performance on the CLEVR dataset shown in Table 11 reveals clear patterns of weakness, primarily centered around analogical and complex relational concepts.

#### B.1.1  ANALOGICAL RELATIONS

The most significant challenge for the VLMs concept is abstract, analogical comparisons. The concept *same size* is a clear example, yielding the lowest F1 score in the entire table for Qwen2VL-7B at 0.604. Ovis1.6-9B also struggles significantly with it, scoring just 0.794. While InternVL2.5-8B performs better, *same color* is a notable weak point for it at 0.840. This indicates a systemic difficulty in performing comparative judgments with visual prompting compared to simple identification.

#### B.1.2  3D SPATIAL RELATIONS ARE A CLOSE SECOND

The next most difficult category is 3D spatial relationships. Concepts like *behind* and *front* consistently score lower than basic attributes across all models. For instance, the F1 scores for *behind* are 0.836 (Ovis1.6), 0.877 (Qwen2VL), and 0.868 (InternVL2.5). This demonstrates a weaker understanding of object relations in three-dimensional space compared to intrinsic properties like shape or material.

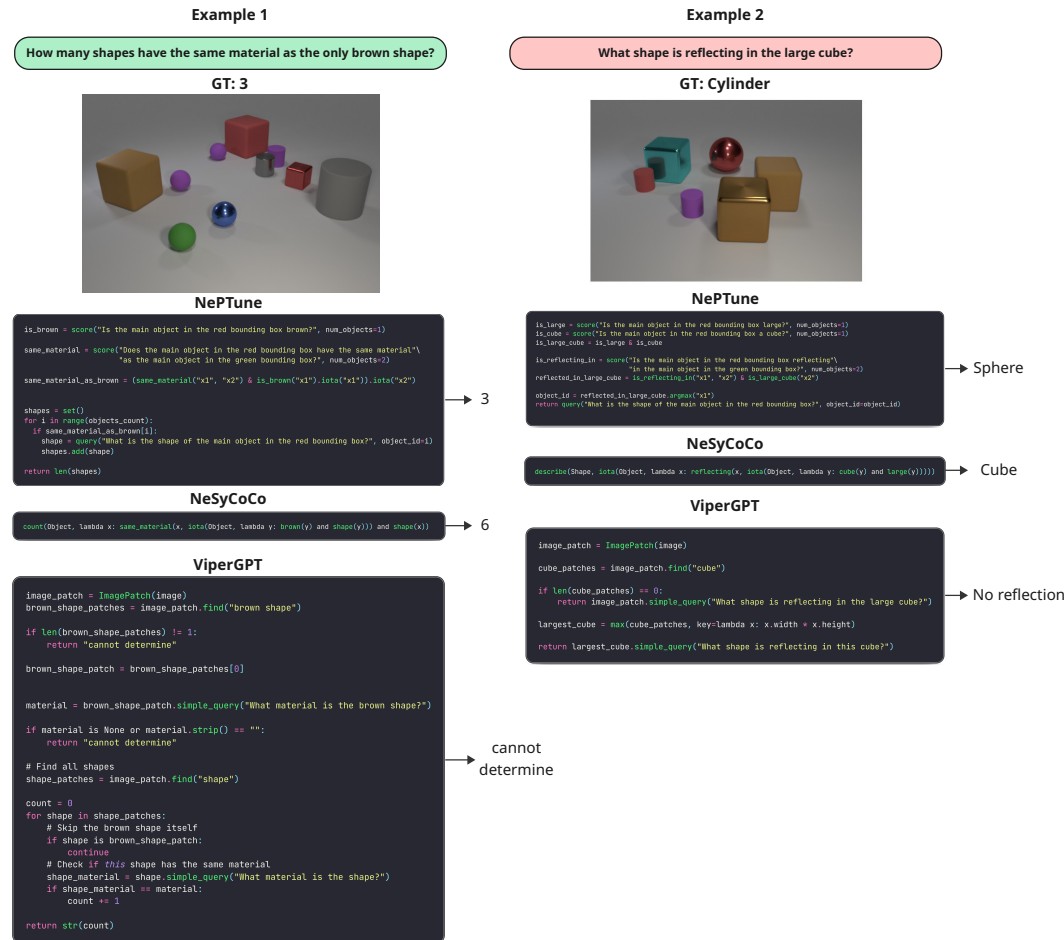

Figure 4: Qualitative examples of NePTune compared to ViperGPT and NeSyCoCo on CLEVR-Humans.

```
is_adult_giraffe = score("Is the main object inside of the red bounding box an adult giraffe?", num_objects=1, type="class")
is_baby_giraffe = score("Is the main object inside of the green bounding box a baby giraffe?", num_objects=2, type="class")
licking = score("Is the adult giraffe inside of the red bounding box licking the baby giraffe inside of the green bounding box?", num_objects=2, type="relation")
final_predicate = is_adult_giraffe('x1') & is_baby_giraffe('x2') & licking('x1', 'x2')
return final_predicate.iota('x1')
```

Figure 5: Qualitative example of wrong NePTune program generation. Mistakes are highlighted with red boxes.

### B.1.3 COLOR CONFUSION

Certain colors also pose challenges, particularly for the Qwen2VL model. The color *cyan* shows a notable performance dip for Qwen2VL (0.805). Further analysis suggests this is due to confusion with visually similar colors like *blue* (0.806) and *green* (0.838). This pattern highlights that while models can identify common colors well, they are less robust with more specific shades and can struggle to differentiate them.

### B.1.4 REAL-WORLD IMAGE CHALLENGES IN VISUAL GENOME

Transitioning from the synthetic CLEVR environment to the complex, real-world images of the Visual Genome (Krishna et al., 2016) dataset reveals a notable shift in performance dynamics. We evaluate our models on atomic queries generated from the cleaned VG scene graphs in GQA (Hudson & Manning, 2019) using these templates:

Table 11: F1 Scores for atomic concept grounding on CLEVR using visual prompting.

| Concept | Ovis1.6 | Qwen2VL | InternVL2.5 |
|---|---|---|---|
| blue | **0.972** | 0.806 | 0.986 |
| brown | **0.989** | 0.921 | **0.989** |
| cube | 0.915 | 0.952 | **0.991** |
| cyan | 0.961 | 0.805 | **1.000** |
| cylinder | 0.964 | 0.960 | **1.000** |
| gray | 0.958 | 0.911 | **0.980** |
| green | **0.989** | 0.838 | 0.988 |
| large | 0.956 | 0.975 | **1.000** |
| metal | 0.961 | 0.965 | **0.997** |
| purple | 0.988 | 0.927 | **1.000** |
| red | 0.973 | 0.829 | **1.000** |
| rubber | 0.980 | 0.924 | **1.000** |
| small | 0.976 | 0.868 | **1.000** |
| sphere | 0.980 | 0.980 | **0.996** |
| yellow | 0.974 | 0.871 | **0.982** |
| behind | 0.836 | 0.877 | **0.868** |
| front | **0.890** | 0.850 | 0.850 |
| left | **0.919** | 0.885 | 0.943 |
| right | **0.934** | 0.863 | 0.930 |
| same_color | **0.982** | 0.957 | 0.840 |
| same_shape | 0.862 | 0.855 | **0.936** |
| same_size | 0.794 | 0.604 | **0.951** |
| **Macro F1** | 0.943 | 0.874 | **0.964** |
| **Micro F1** | 0.903 | 0.866 | **0.945** |

- **Relation Prompt:**
  Is the {class$_1$/object$_1$} in the red bounding box {class$_2$/object$_2$} the {relation} in the green bounding box?

- **Class Prompt:**
  Is the object {inside of/in} the red bounding box {article} {class}?

- **Attribute Prompt:**
  Is the {class} in the red bounding box {attribute}?

As shown in Table 12, a key challenge emerges in fundamental object identification. While the models remain highly proficient at identifying spatial relations and attributes, their performance in the "Class/Object" category is consistently lower than CLEVR. Ovis1.6-9B drops to an F1-score of 0.804, with InternVL2.5-8B and Qwen2VL-7B at 0.818 and 0.833, respectively. This suggests that the visual complexity, clutter, and vast number of fine-grained categories in real-world images make the foundational task of object recognition a more significant hurdle than in the controlled CLEVR environment.

## B.2 End-to-End Performance

These findings align with the paper's broader conclusion that errors in the final reasoning pipeline often originate from the VLM's poor performance on these specific types of relational and analogical concepts. Based on the results of this analysis, in addition to the *score* perceptual interface, we generate a set of the most common spatial predicates, such as *left*, *right*, *behind*, *front*, etc., using the position and depth estimation of bounding boxes for the sake of efficiency and accuracy.

The results in Table 13 clearly show that NePTune enhances the compositional reasoning abilities of strong base VLMs such as Ovis1.6 and InternVL2.5. The most significant gains are on quantitative compositional tasks, with accuracy on counting questions improving by over 12% for InternVL2.5 and counting comparison questions by nearly 11% for Ovis1.6. This highlights the value of our

Table 12: Top 20 common concepts F1-Score comparison on real-world scene graphs.

| Concept | Ovis1.6-9B | Qwen2VL-7B | InternVL2.5 |
|---------|-----------|------------|-------------|
| right of | 0.940 | **0.971** | 0.962 |
| left of | 0.951 | 0.970 | **0.980** |
| man | 0.906 | **0.914** | 0.892 |
| shirt | 0.840 | 0.893 | **0.963** |
| person | 0.821 | 0.792 | **0.824** |
| window | 0.857 | 0.894 | **0.957** |
| pole | 0.842 | 0.900 | **0.927** |
| building | 0.621 | 0.839 | **0.889** |
| tree | 0.750 | 0.833 | **0.839** |
| wall | 0.815 | **0.857** | 0.800 |
| on | 0.636 | **0.727** | 0.696 |
| sign | 0.800 | 0.815 | **0.833** |
| car | 0.870 | 0.960 | **1.000** |
| hand | **0.929** | 0.929 | 0.923 |
| in | 0.444 | 0.421 | **0.500** |
| white | 0.750 | **0.889** | 0.846 |
| near | **0.923** | 0.880 | **0.923** |
| sky | 0.500 | 0.667 | **0.800** |
| ear | 0.818 | 0.800 | **0.909** |
| woman | **0.952** | **0.952** | **0.952** |

Table 13: Comparison of model accuracies across various categories. Maximum values for each NePTune and backbone VLMs are bolded. Changes compared to normal models are marked with colors and arrows.

| Metric | Qwen2VL | Ovis1.6 | InternVL2.5 | Qwen-NeSy | Ovis-Nesy | Intern-Nesy |
|--------|---------|---------|-------------|-----------|-----------|-------------|
| Final Accuracy (%) | **93.55** | 85.00 | 90.25 | 82.55 (↓ 11.00) | 88.80 (↑ 3.80) | **92.65** (↑ 2.40) |
| Exist (%) | **98.21** | 86.74 | 87.10 | 84.59 (↓ 13.62) | 88.53 (↑ 1.79) | **93.19** (↑ 6.09) |
| Query Attribute (%) | 93.77 | 90.14 | **98.26** | 91.30 (↓ 2.47) | 96.23 (↑ 6.09) | **96.81** (↓ 1.45) |
| Compare Attribute (%) | **99.44** | 90.28 | 98.61 | 78.33 (↓ 21.11) | 88.33 (↓ 1.95) | **91.94** (↓ 6.67) |
| Count (%) | **87.50** | 75.40 | 74.60 | 70.16 (↓ 17.34) | 78.83 (↑ 3.43) | **87.10** (↑ 12.50) |
| Compare Number (%) | 90.29 | 78.29 | **90.86** | 88.57 (↓ 1.72) | 89.14 (↑ 10.85) | **92.57** (↑ 1.71) |

structured, programmatic approach for tasks that pure neural models find challenging. Conversely, our framework degrades the performance of the already accurate model Qwen2VL. A deeper look reveals that the primary source of this degradation is the analogical questions, which are the most apparent in the "Compare Attribute" task, where performance drops by 21%. This aligns perfectly with our findings in Experiment 1, which showed that this specific VLM struggles with analogical reasoning (e.g., *same color*, *same shape*). This demonstrates both the power of our framework when paired with a reliable grounding VLM and its sensitivity to the underlying VLM's weaknesses, as errors in perception are propagated by the logical executor.

## C   EXPERIMENTAL SETUP

All experiments were conducted on a server running Ubuntu OS, equipped with an AMD EPYC 7413 24-core CPU, 700GB of system RAM, and an NVIDIA A6000 GPU (48GB). Unless specified otherwise, our core models included DeepSeekV3.1 (DeepSeek-AI-Team, 2025) as the backbone LLM with greedy sampling, Grounding DINO-B for object detection, and DepthAnythingV2 (Yang et al., 2024) for depth estimation. We utilized the Hugging Face Transformers library (Wolf et al., 2020) for VLM interaction and the InternVL-2.5 codebase for finetuning[1]. To supplement the LLM's output, we also used spaCy (Honnibal & Montani, 2017) Named Entity Recognition to extract keywords from the query.

---

[1]https://internvl.readthedocs.io/en/latest/internvl2.5/finetune.html

# D  DATASETS

Here, we provide a detailed breakdown of the datasets used to evaluate the NePTune framework across a range of visual reasoning challenges.

## D.1  CLEVR

We evaluate core compositional reasoning using the **CLEVR (Compositional Language and Elementary Visual Reasoning)** dataset (Johnson et al., 2017a). It contains 3D-rendered synthetic images of simple objects and is used for the task of **Visual Question Answering (VQA)**, where the model must answer complex, programmatically generated questions about object attributes, counts, and relations. We measure performance using **Accuracy (%)** and, for answers requiring semantic equivalence matching (e.g., matching "2" with "two"). For CLEVR dataset benchmarks where a deterministic evaluation did not capture the correctness of the answer, such as matching "2" with "two", we use GPT-4o as an AI judge for evaluation. The prompt used for this evaluation is shown in Figure 7. Since the CLEVR evaluation set was large (700K), we used the first 2000 samples for evaluating our method. We used the best available program generation, similar to Hsu et al. (2024).

### D.1.1  CLEVR QUERY CATEGORIES

Our fine-grained analysis on CLEVR, presented in Table 3, breaks down performance across the following five distinct types of reasoning skills using the programs in the dataset:

**Exist** These questions test for the presence of an object with specific properties, requiring a "Yes/No" answer. For example, *"Is there a large green cube behind the small red sphere?"*

**Query Attribute** These questions ask for a specific property (e.g., color, material) of a uniquely identified object, such as, *"What color is the small shiny cylinder?"*

**Compare Attribute** This category involves "Yes/No" questions that compare a single property between two objects. For example, *"Does the large sphere have the same material as the small cube?"*

**Count** These questions require the model to return the total number of objects matching a description, such as, *"How many red metallic objects are there?"*

**Compare Number** These questions involve comparing the quantities of two different sets of objects, also resulting in a "Yes/No" answer. For example, *"Are there more spheres than cubes?"*

## D.2  CLEVR-HUMANS

The **CLEVR-Humans** dataset (Johnson et al., 2017b) allows us to evaluate performance on more natural language. It uses the same synthetic images as CLEVR but features more complex and linguistically diverse questions written by humans. Similar to CLEVR, performance is measured by Accuracy (%).

## D.3  CLEVR EXTENSIONS

To probe a wider range of reasoning skills, we use a collection of compositional challenges from Hsu et al. (2024), built on the CLEVR environment . Each task tests a unique aspect of complex reasoning:

**CLEVR-Ref** This task tests referring expression grounding, where the model must identify a target object based on its relationship to other objects. The evaluation is conducted by measuring the IOU value greater than 0.5.

> *Example: "There is a sphere that is front the gray cylinder, find the small cylinder that is left of it."*

**CLEVR-Puzzles** This benchmark evaluates the model's ability to solve visual constraint satisfaction problems by finding a set of objects that simultaneously satisfy multiple attribute and relational constraints.

> *Example: "Can you find four objects from the image such that: object 1 is a large metal object; object 2 is a metal object; object 3 is a small rubber cylinder; object 4 is a small yellow metal cylinder; object 1 is front object 2; object 1 is behind object 4; object 1 is behind object 3."*

**CLEVR-RPM** This task tests abstract relational reasoning by mimicking Raven's Progressive Matrices. The model is presented with objects in a grid that follow a pattern and must identify a candidate object from the scene that correctly completes that pattern.

> *Example: "There are 9 objects, ordered in a 3x3 grid: row 1 col 1 is a small rubber object; row 1 col 2 is a small rubber object; row 1 col 3 is a large rubber object; row 2 col 1 is a small metal object; row 2 col 2 is a small metal object; row 2 col 3 is a large metal object; row 3 col 1 is a small metal object; row 3 col 2 is a small metal object; I am missing one object at row 2 col 2. Can you find an object in the scene that can fit there?"*

## D.4 REF-ADVERSARIAL (REF-ADV)

To test performance in real-world scenarios, we use the **Ref-Adversarial** benchmark (Akula et al., 2020). This dataset consists of real-world images with visually similar distractor objects, making the task of Referring Expression Grounding particularly challenging. REG requires the model to locate the specific object corresponding to a text description, and we evaluate the model using Grounding Accuracy (%), where a prediction is considered correct if the Intersection over Union (IoU) with the ground-truth bounding box is 0.5 or greater.

## D.5 REF-GTA

To measure generalization under a significant domain shift, we use the **Ref-GTA** benchmark (Tanaka et al., 2019). This is a REG benchmark where images are sourced from a photo-realistic game simulation, creating an out-of-distribution challenge for models pre-trained on natural images. Performance is measured by Grounding Accuracy (%), defined by an Intersection over Union (IoU) threshold of 0.5 between the predicted and ground-truth bounding boxes.

## E VERIFICATION

> You're an image analyst designed to check if the highlighted objects in the image meet the query description, and which one is more likely to meet the query description.
> The query is: "{query}"
> Please check the highlighted object "0" [in the red bounding box] and "1" [in the green bounding box] in the image and answer the question: Which object is more likely to meet the query description? Your answer should be "0", "1". Answer with one word or phrase.

Figure 6: Prompt used for the Pairwise Arbiter verification strategy.

To recover from occasional symbolic execution failures, we add a lightweight *verification* stage that operates after reasoning has produced a candidate answer. We explore two simple strategies:

1. **Pairwise Arbiter:** The model is presented with two candidates, the backbone prediction and the symbolic reasoning output, along with the query. It acts as an arbiter, directly judging which option better satisfies the description. Our prompt, similar to the one used by NAVER for this process, is shown in Figure 6.

2. **Confidence Gating:** We compute a softmax distribution over the symbolic executor's scores with temperature $T$, and use a threshold $\tau$ to decide whether to trust the executor. If $\max(\text{softmax}) < \tau$, we fallback to the backbone prediction; otherwise, we keep the executor's output. Both $\tau$ and $T$ are tuned on a 500-example held-out set for each backbone.

Table 14 summarizes the performance of both light-weight verification strategies across different backbones. Overall, we observe that the *Pairwise Arbiter* tends to outperform the *Confidence Gating* method, highlighting the benefit of leveraging a VLM to arbitrate between backbone and symbolic predictions.

Table 14: Post-verification accuracy on Ref-Adv.

| Backbone VLM | Verification | Accuracy (%) | $\tau$ | $T$ | Symbolic Share(%) |
|---|---|---|---|---|---|
| **Qwen2VL-7B** | Confidence Gating | 80.91 | 0.70 | 0.40 | 73.59 |
| | Pairwise Arbiter | 80.93 | N/A | N/A | 80.23 |
| **Ovis1.6-9B** | Confidence Gating | 60.49 | 0.30 | 0.10 | 86.79 |
| | Pairwise Arbiter | 63.25 | N/A | N/A | 29.73 |
| **InternVL2.5-8B** | Confidence Gating | 76.65 | 0.60 | 0.50 | 71.06 |
| | Pairwise Arbiter | 78.08 | N/A | N/A | 77.86 |

> You are an automatic answer checker! I will give you a question and answer, and a generated response, and you will tell me if the response is correct given the ground truth answer. If the response is correct, you will say <Yes>, otherwise you will say <No>. Rubber and Matte are the same. Shiny and Metal or Metallic are the same. Square and Cube are the same. Yellow and Golden colors are similar in the images, too. Balls and spheres are similar. Check if the response is correct given the questions and the ground truth answer, deeply thinking about the context. Answer with <Yes> or <No> and don't mention them other than for the final answer. Important! Ignore the red bounding box reference in the generated response.
> Question: "{Question}"
> Ground Truth Answer: "{Answer}"
> Generated Response: "{Prediction}"

Figure 7: Visual Question Answering judge LLM prompt.

## F    Hyperparameters

The fine-tuning hyperparameters for our experiments are detailed in Table 15. These settings are based on the configurations defined in our training script.

During our experiments, we found that applying LoRA fine-tuning to the vision backbone, in addition to the language model, was highly effective for domain adaptation. To illustrate, without any neuro-symbolic fine-tuning, the base NePTune framework achieved **56.49%** accuracy on Ref-GTA. In contrast, a standard fine-tuning approach on the base VLM only reached **4.19%** accuracy. This performance gap underscores the effectiveness of our neuro-symbolic method and justifies fine-tuning the vision components to achieve optimal results in novel environments.

## G    Latency Analysis

The primary computational cost of NePTune stems from the calls to the VLM for concept grounding. For a scene with $N$ candidate objects, $R$ binary relations, and $C$ unary concepts, scoring a unary concept (such shape, attribute) requires $N$ VLM calls (overall $O(CN)$ across all unary concepts), and scoring a binary relation requires $N^2$ VLM calls (overall $O(RN^2)$ across all relations used). This can be slow for scenes with many candidate objects. To reduce this computational overhead, we apply a simple optimization. Before execution, we collect the set of required atomic concepts from the generated program and issue them in a single batched call, with prefix caching enabled, using the vLLM (Kwon et al., 2023) tool. Since the image tokens and most textual prefixes are shared across these queries, prefix caching eliminates redundant computations.

Table 15: Hyperparameters for Fine-tuning Experiments

| Hyperparameter | Standard Fine-Tuning | NePTune Neuro-Symbolic Fine-Tuning on Ref-Adv | on Ref-GTA |
|---|---|---|---|
| ***General Training*** | | | |
| Learning Rate (lr) | $4 \times 10^{-5}$ | $4 \times 10^{-5}$ | $4 \times 10^{-5}$ |
| Batch Size | 1 | 1 | 1 |
| Gradient Accumulation | 4 | 4 | 4 |
| Weight Decay | 0.01 | 0.01 | 0.01 |
| LR Scheduler | Cosine | Cosine | Cosine |
| Warmup Ratio | 0.03 | 0.03 | 0.03 |
| Loss Function | Cross-Entropy | Binary Cross-Entropy | Binary Cross-Entropy |
| ***LoRA Configuration*** | | | |
| Rank (r) | 16 (LM), 8 (Vision) | 16 (LM) | 16 (LM), 8 (Vision) |
| Alpha ($\alpha$) | 32 (LM), 16 (Vision) | 32 (LM) | 32 (LM), 16 (Vision) |
| Dropout | 0.05 | 0.05 | 0.05 |

Figure 8 summarizes the average latency of NePTune on Ref-Adv, using the experimental setup described in Appendix C, on a single GPU.

- **For InternVL2.5-1B:** Grounding with a VLM forward pass takes 0.90 s, while the average non-cached NePTune query takes 3.63 s. Our caching optimization reduces this average to 1.30 s, revealing a small 0.4 s overhead.

- **For InternVL2.5-8B:** Grounding with a VLM forward pass takes 1.48 s, the non-cached average is 12.35 s, and the cached average drops to 8.19 s.

In other words, NePTune with caching adds 0.4 s of overhead for the 1B backbone and 6.7 s for the 8B backbone, remaining a small multiple of a VLM forward pass.

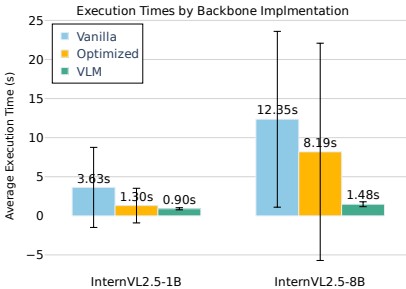

Figure 8: Latency Analysis of NePTune over Ref-Adv dataset. "Vanilla" refers to non-cached execution, "Optimized" is our optimized approach, and "VLM" is a single forward pass.

The latency averages reported in Figure 8 are heavily affected by a few computationally intensive queries. Figure 9 shows the full execution time distribution for both models. For the **1B model**, the vast majority of queries are extremely fast, with 3,030 queries completing in **under 2 seconds**. For the **8B model**, most queries (2,143) are resolved in **under 5 seconds**. However, we also observe a longer tail, with 229 complex queries taking longer than 25 seconds, which increases the average.

Figure 10 provides a detailed breakdown of *why* certain queries are slow, plotting time as a function of the **Number of Proposal Objects** (x-axis, $N$) and the **Number of Concepts** (y-axis, $K = R+C$) for both InternVL2.5-1B and 8B.

These heatmaps confirm that latency increases with both $N$ and $K$, but the strongest spikes correlate with large $N$ in relational predicates, which matches the $O(RN^2)$ cost of relational scoring. For the 1B backbone, average latency increases from about 0.43 s at $N = 2$ to about 2.56 s at $N = 6$. For the 8B backbone, average latency increases from about 2.51 s at $N = 2$ to about 16.70 s at $N = 6$.

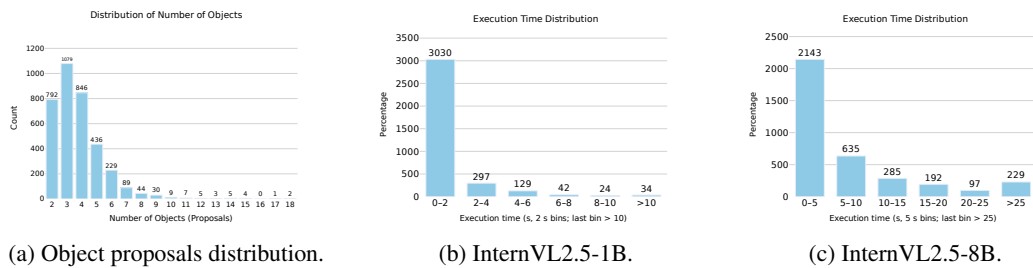

(a) Object proposals distribution.    (b) InternVL2.5-1B.    (c) InternVL2.5-8B.

Figure 9: Number of object proposals (a) and latency distribution for InternVL2.5-1B (b) and InternVL2.5-8B (c) on Ref-Adv.

As shown in Figure 9a, on Ref-Adv, roughly 90% of queries have $N \leq 6$, so most points lie in this low to moderate regime. Only about 1% of queries have $N \geq 10$, and these high-$N$ relational cases account for the few cells that reach very large runtimes, including a maximum of about 257 s at $(K = 6, N = 18)$ for the 8B model.

This analysis shows that NePTune introduces a controlled overhead compared to a single VLM pass, with large costs concentrated in rare queries that involve many proposal objects and multiple relational concepts. We view this as a favorable trade-off given the substantial gains in compositional robustness and NePTune's ability to solve complex queries that the backbone models alone fail to answer.

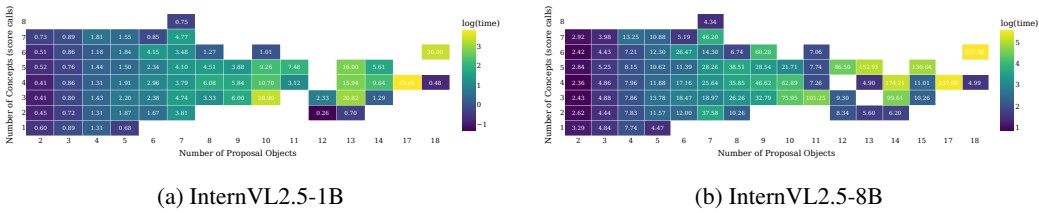

(a) InternVL2.5-1B    (b) InternVL2.5-8B

Figure 10: Heatmap of time for InternVL2.5-1B (a) and InternVL2.5-8B (b). Bright yellow cells indicate high-latency queries, which correlate strongly with a high number of proposal objects. Cells show the average execution time of the bin in seconds and are colored by $\log(\text{time})$ for better visualization.

## H HUMAN EVALUATION OF PROGRAM GENERATION

In this section, we quantify how often NePTune's LLM program generator produces incorrect programs and what kinds of errors it makes, using a small but carefully controlled human evaluation across three benchmarks: Ref-Adv, Ref-GTA, and CLEVR-Humans.

### H.1 EXPERIMENTAL SETUP

For each dataset, we randomly sampled 200 questions from the corresponding evaluation sets used in our main experiments. For each question, we ran NePTune's program generator and obtained a Python program. This program, together with the question and its output, was then evaluated by human raters. We used three evaluators with knowledge of machine learning and Python programming for each dataset. Before starting evaluation, we held a session in which we explained the NePTune framework (bounding boxes, `score` and `query` calls, soft logic operators) and walked through several example questions and their programs.

Each evaluator then independently labeled all 200 programs for each dataset. During the evaluation, one of the authors was available online to answer questions about ambiguities and border cases for the evaluators.

Table 16: Program evaluation outcomes per dataset (200 questions each). The **Correct** column shows the number of programs marked *correct* by all evaluators; the error columns count programs marked with that error by at least one evaluator; and $\kappa$ is the average pairwise Cohen's $\kappa$ for the binary correctness decision.

| Dataset | Correct | Syntax | Atomic | Logic | Other | Accuracy | $\kappa$ |
|---|---|---|---|---|---|---|---|
| Ref-Adv | 175 | 0 | 21 | 3 | 1 | 87.50% | 0.76 |
| Ref-GTA | 187 | 0 | 12 | 1 | 1 | 93.50% | 0.73 |
| CLEVR-Humans | 195 | 0 | 0 | 5 | 0 | 97.50% | 0.71 |

## H.2 EVALUATION SCHEME

For each question and generated program, evaluators followed a two-step decision rule.

**Step 1: correctness.** Evaluators first decided whether the program is *fully correct*:

- **Mark as correct.** The program is syntactically valid and logically faithful to the question and implements the intended reasoning steps.

**Step 2: error tags.** If the program was *not* marked as correct, evaluators selected from the following error labels that applied:

- **Syntax error.** The program has a Python syntax error or fails at runtime due to an obvious coding mistake, for example, a wrong number of arguments to `score`, a misspelled function name, or an indexing error.

- **Wrong atomic question.** One or more atomic predicates in the program are incorrectly decomposed from the natural language query. This includes mis-specifying the question passed to `score` or `query`. For example, dropping or adding attributes, using singular instead of plural (or vice versa), selecting the wrong bounding-box reference or color, or assigning the wrong arity (using a unary predicate where a binary relation is required). These errors reflect a mismatch between the intended atomic meaning of the query and the atomic predicates used in the program. For instance, if the query is talking about "two men sitting on a table" but the code decomposes that to score("... a man?") or the query is asking about "dining table" but the atomic query in decomposition just says "table".

- **Wrong logic.** The program executes but implements the wrong logical structure for the question (for example, using a disjunction instead of a conjunction, mis-handling quantifiers such as "exactly one", or applying a condition to the wrong set of objects).

- **Other issue.** Any remaining problem that does not fit the above categories, including ambiguous cases or issues clearly caused by the underlying VLM output rather than by the program structure.

## H.3 LABEL DISTRIBUTION AND AGREEMENT

Table 16 summarizes the label distribution across evaluators for each dataset. For each dataset, we report the number of programs marked at least once with each error label (union over the three evaluators), the accuracy (fraction of programs marked *correct* by all evaluators), and the average pairwise Cohen's $\kappa$ computed for the binary correctness decision (*Correct* vs. *Incorrect*). Inter-evaluator agreement, measured as average pairwise Cohen's $\kappa$, was consistently high across datasets.

## H.4 DATASET SPECIFIC ERROR PATTERNS

The human study also reveals qualitatively different error patterns across the three datasets.

**CLEVR-Humans.** Typical issues include using *iota* normalization for indefinite descriptions, or miscounting when the reference object should be excluded from a set and subtle logical composition.

In these cases, the atomic score calls are usually reasonable, but the higher-level program does not faithfully capture uniqueness and distinctness. For instance, to answer "Is there a small red sphere next to a large red sphere", the program applies `small_red_sphere.iota("x1")`, which can lead to a wrong answer if there is no red sphere. As we expected, CLEVR-Humans is a dataset with a simple closed set of concepts; therefore, wrong atomic queries didn't exist.

**Ref-Adv and Ref-GTA.**    In these two datasets, the main failure mode concerns *atomic prompting* for plural and group expressions. Many referring expressions in these datasets refer to groups of objects (e.g., "the children in the background" versus "the child on the left"). Our evaluators found that while the overall program structure is usually correct, the generator often translates these into atomic prompts that implicitly assume a single object or drop the plurality constraint (e.g., asking about "the child" inside `score` when the query is about "children"). This may lead to wrong region selection even if the tensor logic and local perception are otherwise sound. We also identified a subset of atomic prompting errors involving simple template issues, specifically red/green bounding box mismatches. Because these followed consistent patterns, we were able to detect and fix them automatically using rule-based pattern matching. Finally, cases marked with *other* generally concern ambiguity. For instance, given the expression *"The telephone closest to the screen,"* the reference to "screen" is unclear; it could mean the camera viewpoint or a physical screen in the image. Using images alongside the query in future work might help mitigate this issue. Together with the low number of syntax and logic errors, this suggests that NePTune's main remaining bottlenecks lie in subtle specification mismatches (especially plurality) rather than gross program failures.

## H.5    SUMMARY

Overall, the human study shows that NePTune's program generator is reliable in practice: most programs are judged correct, syntax errors are rare, and inter-evaluator agreement is high. The remaining issues are localized specification errors rather than gross logical failures. On CLEVR-Humans, using the normalization function *iota* was the most common error, while on Ref-Adv and Ref-GTA, the dominant errors come from atomic prompts that drop plural or group cues in the natural language passed to `score`/`query`. Several of these atomic prompt bugs follow simple patterns that can be addressed with some prompt tuning.

