# OpenReview forum: "NePTune: A Neuro-Pythonic Framework for Tunable Compositional Reasoning on Vision-Language"
_ICLR.cc/2026/Conference — ICLR 2026 Poster_

### Official Review · Reviewer_vXFY · 2025-10-21

**Soundness:** 3
**Presentation:** 3
**Contribution:** 3
**Rating:** 8
**Confidence:** 4

**Summary:**

This paper investigates integrating both the coding and perception abilities of foundation models within a near-symbolic framework to improve the reasoning failures of the foundation models alone. The authors propose a three stage processing pipeline including structured and targeted queries to VLMs which are integrated within code executions obtained from an LLM. The goal of the approach is to tackle zero-shot processing. The authors evaluate their Neptune approach on several, mainly CLEVR-based datasets.

**Strengths:**

The topic of this work tackles many relevant aspects of current foundation models. The experimental evaluation is generally well structured and presented. The results show promising boosts particularly over the pure VLM/LLM baselines.

**Weaknesses:**

Generally, I don't have any fundamental issues, just some information is missing and a little unclear. I think if the authors could clarify these/ integrate them into the paper, it would be easier to understand the contribution of the work.

What I am definitely missing is a section specifying the exact model setups and high-level implementation information regarding Neptune, but also the baselines. I would place this before section 4.1. I.e. what models were used for the Neptune results? What models are being compared to? Currently all of this information is spread out, if at all, across section 4.

The authors provide good RQs in Section 4, however these are never referenced after being introduced. For improved readability I suggest to reference them within each experimental subsection.

I am a little confused about which baselines are compared to for which evaluations. E.g. why is MDETR used in Table 5, but not for the evaluations for Table 3 or 4? Same goes for Naver.

Why don't the authors discuss and compare to older NeSy approaches such as [1]?

I find the Discussion particularly around table 8 somewhat confusing. What are the authors trying to show? In my understanding the point is to show that the basic "perception" performance are quite high?

[1] Yi, Kexin, et al. "Neural-symbolic vqa: Disentangling reasoning from vision and language understanding." Advances in neural information processing systems 31 (2018).

**Questions:**

What exactly is a "response" in ll 236?

How does Neptune decide when to use a "query"

What is the intuition/benefit of having soft compositional reasoning components? The authors seemed quite brief on this.

I don't quite understand the intuition/explanation for the weaker "query attribute" and "compare attribute" results in table 3? Could the authors please elaborate on this?

Tab. 7 how is the VLM fine-tuned? What for is it fine-tuned?

Why not integrate table 9 with the other evaluations around RQ 1?

What is the importance of repeatedly naming models end-to-end when they are not actually trained end-to-end for the evaluations in the paper? Moreover, if Neptune is based on these end-to-end models, also Neptune can profit from end-to-end pretraining.

---

> ### Author Response · Authors · 2025-11-25
> **Response to Reviewer vXFY (1/2)**
>
> We thank our reviewer for their strong support and actionable suggestions for improving the paper's clarity.
>
> - **W1–W3:** (presentation: models, baselines, and RQ references).
>
> In the updated version, we clarify the experimental setup at the start of Section 4 (4.1), providing a brief “experiments at a glance” overview that maps each RQ to the evaluation datasets, backbone VLMs, and baseline families used. We also explicitly point to Table 9 there as a compact summary of the backbone and NePTune across benchmarks, and we update each experimental subsection to reference the corresponding RQ.
>
> > **W4:** Confusing baselines (MDETR, Naver).
>
> We chose our baselines based on the goal of each experiment and their competitiveness:
>
> - **MDETR** is a fully supervised neural model trained on CLEVR. This baseline was used in earlier work to compare NeSyCoCo and LEFT, so we included it on CLEVR-Humans (Table 5) as a representative of strong trained neural baseline. Trained neural methods handle linguistic variety well (common in human-generated queries), where symbolic methods are known to struggle, as reflected in the LEFT and NeSyCoCo results. However, MDETR is essentially saturated on synthetic CLEVR and adds little insight.
>
>  - **NAVER** is designed specifically for the referring expression grounding task. We therefore include it only in the REG experiments on Ref-Adv and Ref-GTA (Tables 6 and 7), where the task exactly matches its design.
>
> We now explicitly clarify these baseline choices in Section 4.1, making it clear which baselines are relevant in each setting.
>
> > **W5:** Why don't the authors discuss and compare to older NeSy approaches such as Yi et al.?
>
> NS-VQA is an important early neuro-symbolic VQA system. In the submission version, we compared with LEFT, which is more recent and versatile than NS-VQA. For a more comprehensive positioning, we have added this comparison to the related work section.
>
> > **W6:** I find the Discussion, particularly around table 8, somewhat confusing. What are the authors trying to show? In my understanding the point is to show that the basic "perception" performance are quite high?
>
> Yes, the goal of Table 8 is to show that basic “perception” performance on atomic predicates is already quite high. In that table, we probe the VLMs only on atomic predicates (single attributes, objects, and relations) using simple questions. We find that raw perception is strong; for example, InternVL2.5 achieves 0.945 F1 on CLEVR atomic predicates, while its accuracy as a monolithic model on CLEVR-Ref is only 27%. This gap shows that most failures on compositional benchmarks stem from reasoning over these atomic scores rather than from recognizing basic concepts. This experiment supports our decision to use VLMs with visual prompting for atomic concept grounding.
>
> > **Q1:** What exactly is a "response" in l. 236?
>
> The “response” in line 236 is the (typically single-token) answer generated by the VLM to the atomic question passed to the `query` function. It is simply the most probable token returned by the VLM. We clarified this notation in the revised text.
>
> > **Q2:** How does NePTune decide when to use a "query"?
>
> The LLM that generates the program makes the choice based on the few-shot examples in the prompt. The code semantic evaluation shows that the model generates these queries with high accuracy given the questions.
>
> > **Q3:** What is the intuition or benefit of having soft compositional reasoning components?
>
> The main benefit is handling uncertainty at the perception level. NePTune never collapses VLM scores into hard true/false decisions at the perception level. Instead, it keeps the uncertainty scores from the VLM and combines them with soft operators (for example, min for AND and max for OR), so the symbolic reasoner can still pick the most plausible configuration even when individual predicates are uncertain. In addition, our differentiable soft composition functions enable tuning the backbone model within the symbolic program.
>
> **Example of soft composition.** Suppose there are two objects and we are looking for a “red circle”. The VLM gives red scores `[0.1, 0.4]` and circle scores `[0.6, 0.9]`. A crisp logical system with a threshold of 0.5 would require both `red[i] > 0.5` and `circle[i] > 0.5` for some object `i`, and would fail here because no object passes both thresholds.
>
> In NePTune, we keep the scores and form a soft conjunction per object, for example
> $(\text{red} \wedge \text{circle})_i = \min(\text{red}_i, \text{circle}_i) = [0.1, 0.4],$
> and then select the object with the highest combined score. This lets the system make a global decision about the best candidate rather than discarding uncertain but useful evidence.

---

> > ### Author Response · Authors · 2025-11-25
> > **Response to Reviewer vXFY (2/2)**
> >
> > > **Q4:** I don't quite understand the intuition/explanation for the weaker "query attribute" and "compare attribute" results in table 3? Could the authors please elaborate on this?
> >
> > Our further error analysis (Appendix B) reveals that the main source of error in the `compare attribute` is atomic perceptual grounding for analogical relations (eg, same_color, same_shape), which dominate the `compare attribute` questions.
> >
> > For instance, Table 12 shows that `same color` reaches only 0.84 F1. The queries in the compare-attribute category all involve these analogical predicates (for example, "Do these small cubes have the same color?"), So the lower performance there is driven by VLM perception rather than by the symbolic executor.
> >
> > > **Q5:** Tab. 7 how is the VLM fine-tuned? What for is it fine-tuned?
> >
> > **NePTune + Fine-tuning** in Table 7:  It updates **only** the LoRA weights attached to the VLM backbone. For each referring expression, the NePTune program returns a score vector \(X\), where \(x_i\) is the predicted probability that candidate box \(i\) is the target. We backpropagate the cross-entropy loss over this vector, encouraging the ground-truth box to have the highest probability. In other words, the VLM is fine-tuned to produce better atomic scores.
> >
> > **InterVL + Fine-tuning**: For instruction fine-tuning of the VLM for referring expression, we used the [internvl instructions](https://internvl.readthedocs.io/en/latest/internvl2.0/finetune.html)  to train the VLM to output coordinates in a single pass.
> >
> > > **Q6:** Why not integrate table 9 with the other evaluations around RQ1?
> >
> > The main reason for the separation is that Table 9 goes beyond RQ1. The discussion around Table 9 reports NePTune's performance across multiple benchmarks, demonstrating its ability to improve any model. For that reason, we keep Table 9 in the discussion section, where we analyze cross-RQ trends, and refer to it as a "backbones at a glance" summary instead of treating it as part of the RQ1-specific CLEVR results.
> >
> > > **Q7:** What is the importance of repeatedly naming models end-to-end when they are not actually trained end-to-end for the evaluations in the paper? Moreover, if Neptune is based on these end-to-end models, also Neptune can profit from end-to-end pretraining.
> >
> > In the current draft, we use "end-to-end" too broadly to refer to both how these VLMs and neural methods are pretrained and how they are used as direct prompting answerers in our baselines.
> >
> > To avoid confusion, we revised the wording in the paper to replace "end-to-end" with more precise terms, such as "direct VLM prompting" or "pure neural models," when we refer to the inference-time behavior of the backbones. NePTune builds on the same pretrained VLMs as the direct baselines and fully benefits from their pretraining.  Indeed, our contribution is orthogonal to the pretraining procedure and lies in how these backbones are used at inference time, or can be trained more effectively via a symbolic executor that greatly improves compositional reasoning and robustness.

---

### Official Review · Reviewer_fRHk · 2025-10-28

**Soundness:** 3
**Presentation:** 3
**Contribution:** 4
**Rating:** 6
**Confidence:** 4

**Summary:**

The paper presents NePTune, a neuro-symbolic framework aimed at enhancing compositional reasoning with Vision-Language Models (VLMs). NePTune consists of three main components: an LLM-based program generator, a perceptual grounding module, and a symbolic executor. The program generator converts natural language queries into executable Python programs that can query a VLM to extract object-specific information. The resulting perceptual outputs are then combined using soft logic operators, allowing the system to reason over uncertain predictions from the VLM. The framework operates in a training-free manner but remains differentiable for fine-tuning.

**Strengths:**

- The paper introduces a novel neuro-symbolic design that effectively integrates the power of VLMs, symbol grounding, and explicit compositional reasoning under uncertainty.
- The score-based perceptual grounding interface elegantly connects neural perception with symbolic reasoning.
- The modular, differentiable design is appealing as it supports both zero-shot use and potential fine-tuning without dataset-specific supervision.

**Weaknesses:**

- While the breadth of experiments is commendable, the sheer number of evaluations can make the presentation somewhat overwhelming. In particular, the new experiments introduced in the discussion section are confusing, as they could be more appropriately integrated into the main experimental section. For example, Table 9 provides a helpful overview of the different datasets and could be placed earlier to illustrate NePTune’s general effectiveness. More broadly, adding a concise overview of the datasets, models, and baselines at the beginning of the experimental section would substantially improve readability. It might also be worth considering focusing on a smaller subset of key experiments in the main paper and moving some of the less central results to the appendix. For instance, Table 5 appears useful as supplementary evidence but is not discussed in much detail, suggesting it could be moved without loss of clarity.

- The paper would benefit from a more detailed discussion of how the framework scales when applied to larger VLMs or more complex perceptual inputs, as well as an analysis of its computational overhead compared to direct prompting. It would also be valuable to elaborate on potential limitations that may arise despite high perceptual quality in the VLM or strong program generation capabilities from the LLM.

**Questions:**

- Q1: Is the score function currently limited to pairs of objects, or can it be extended to higher-arity relations?
- Q2: What is the number of tokens that the query function can return?
- Q3: Line 241/242 seems to have a missing or incomplete expression "…’).". Could you clarify?
- Q4: For program generation (component 1), do you use greedy decoding or a sampling-based strategy? Same question for the baseline VLMs
- Q5: Regarding Table 4, could you clarify what "ground-truth programs" refers to, how they are obtained, and why the dataset "Puzzles" can not benefit as strongly from it?
- Q6: What motivated the choice of VLMs for NePTune and comparisons? Do you expect similar zero-shot results for larger models (e.g., >70B or GPT-5 etc.) as in Tables 5 and 6?
- Q7: Can you estimate the computational or token cost of NePTune compared to direct prompting? Given the small amount of tokens for the score and query functions, it seems low, but quantitative insight or discussions would be helpful.

Generally, I like the proposed method and am willing to raise my score if my concerns and questions are addressed.

---

> ### Author Response · Authors · 2025-11-24
> **Response to Reviewer fRHk (1/2)**
>
> We appreciate suggestions for improving the presentation and address them below, along with the specific questions.
>
>
> > **W1**: While the breadth of experiments is commendable, the sheer number of evaluations can make the presentation somewhat overwhelming. In particular, the new experiments introduced in the discussion section are confusing, as they could be more appropriately integrated into the main experimental section. For example, Table 9 provides a helpful overview of the different datasets and could be placed earlier to illustrate NePTune's general effectiveness. More broadly, adding a concise summary of the datasets, models, and baselines at the beginning of the experimental section would substantially improve readability. It might also be worth considering focusing on a smaller subset of key experiments in the main paper and moving some of the less central results to the appendix. For instance, Table 5 appears useful as supplementary evidence but is not discussed in much detail, suggesting it could be moved without loss of clarity.
>
>
> In a revised version, we made the experimental section easier to navigate with small presentation changes. First, we add an overview in a new subsection for the experimental setting, explaining the datasets, backbone VLMs, and baseline families, so readers can see the overall structure of the experiments before any numbers.
> Second, we pointed explicitly to the overview Table at the start of Section 4 as the "experiments at a glance" summary of datasets and tasks. These clarifications should make the experimental narrative more straightforward to follow. We also integrated Tables 5 and 10 to tighten our experiments on CLEVR-Humans.
>
> > **W2:** The paper would benefit from a more detailed discussion of how the framework scales when applied to larger VLMs or more complex perceptual inputs, as well as an analysis of its computational overhead compared to direct prompting. It would also be valuable to elaborate on potential limitations that may arise despite high perceptual quality in the VLM or strong program generation capabilities from the LLM.
>
> We now provide a quantitative latency analysis in **Appendix G**. On Ref-Adv, NePTune is only a small multiple slower than a single VLM call: about **1.3 s per query for the 1B backbone** (81% of queries under 2 s) and **8.2 s for the 8B backbone** (57% under 5 s). The main scaling bottleneck is the number of candidate objects $N$, but in our benchmarks $N$ is small (average $N \approx 3.6$), so only rare scenes with many objects cause noticeable overhead. Even with strong VLMs and LLMs, NePTune can still fail when queries are ambiguous and require additional visual information to disambiguate the query during program generation. You can refer to General responses for more details on latency and program generation quality.
>
>
> > **Q1:** Is the score function limited to pairs, or can it be extended to higher-arity relations?
>
>
> Yes. While NePTune supports higher-arity relations, we limited our program generation to unary and binary in the prompt instructions to reduce computational cost. The higher arity means more object candidate combinations, leading to more VLM calls.
>
>
>
> > **Q2:** What is the number of tokens that the query function can return?
>
> The `query(x)` function imposes a hard 10-token limit, but it is designed to produce shorter answers. We append the instruction `x` with `\nAnswer the question using a single word or short phrase.`, and in practice, the VLM almost always returns a single token (for example, "red", "sphere"). We chose this instruction because it matches common instruction-tuning prompts for these VLMs.
>
> > **Q3:** Line 241/242 missing expression.
>
> We meant to show the expression partially. However, we see that this causes confusion. In the revised version, we have removed that broken line and replaced it with a minimal example that matches our implementation, where `query` is used to retrieve attribute values as intermediate variables in multi-step reasoning.
>
>
> > **Q4:** For program generation (component 1), do you use greedy decoding or a sampling-based strategy? Same question for the baseline VLMs?
>
> We use greedy decoding for all program generation and VLMs. We added this information in Appendix C.

---

> > ### Author Response · Authors · 2025-11-24
> > **Response to Reviewer fRHk (2/2)**
> >
> > > **Q5:** Regarding Table 4, could you clarify what "ground-truth programs" refers to, how they are obtained, and why the dataset "Puzzles" can not benefit as strongly from it?
> >
> > By "ground-truth programs," we refer to the gold symbolic programs released with the synthetic CLEVR-based benchmarks (Ref, Puzzles, RPM) in the LEFT codebase. For the $\dagger$ rows in Table 4, we bypass the LLM parser and execute these provided programs directly with NePTune's executor and concept scores, i.e., we follow the exact reasoning logic specified by the dataset authors.
> >
> > Although CLEVR-Puzzles provides gold programs, performance is still limited by visual grounding rather than reasoning. These queries involve more complex spatial relations such as “behind” and “in front of”, which are challenging even for strong VLMs, as shown in Appendix B.1. As a result, having the correct symbolic program helps less than in datasets where the main bottleneck is program generation.
> >
> >
> > > **Q6:** What motivated the choice of VLMs for NePTune and comparisons? Do you expect similar zero-shot results for larger models (e.g., >70B or GPT-5 etc.) as in Tables 5 and 6?
> >
> > We chose these VLMs because they were state-of-the-art, open-source, and efficient enough to support repeated calls at scale. We expect our findings to transfer and likely improve with larger or stronger VLMs, especially in novel situations not included in their training.
> >
> >
> > > **Q7:** Can you estimate the computational or token cost of NePTune compared to direct prompting? Given the small amount of tokens for the score and query functions, it seems low, but quantitative insight or discussions would be helpful.?
> >
> > Due to the multiple calls to the VLM, we have computational overhead and additional cost compared to running the query on the VLM in a single pass. We use prefix caching to reduce the overall token usage. We provide more details on the latency and computational cost of NePTune in Appendix G and summarize the results in General Response 2.

---

> > > ### Comment · Reviewer_fRHk · 2025-11-27
> > >
> > > Dear authors,
> > > Thank you for the clarifications and additional insights. I appreciate the updates to the experimental section and think it improves clarity. I like the method, especially the smooth integration of symbolic (fuzzy) logic with VLM-based query function. I will raise my score to 8.

---

### Official Review · Reviewer_9EoX · 2025-10-30

**Soundness:** 3
**Presentation:** 3
**Contribution:** 3
**Rating:** 6
**Confidence:** 3

**Summary:**

This paper introduces NePTune, a neuro-symbolic framework for vision-language (VL) reasoning that addresses limitations in prior work, such as brittle pipelines and a lack of adaptability. NePTune translates a natural language query into an executable Python program, similar to models like ViperGPT. However, it makes two key contributions:

- Hybrid Execution Model: The generated Python program combines standard imperative control flow (loops, conditionals) with "soft logic" operators (based on fuzzy logic, e.g., t-norms for AND) that operate directly on the continuous uncertainty scores provided by a Vision-Language Model (VLM).

- Differentiability: These soft logic operations are differentiable, allowing the entire framework (specifically the VLM backbone) to be fine-tuned via "neuro-symbolic fine-tuning" for domain adaptation.

Operating in a training-free manner, NePTune already outperforms strong baselines. When fine-tuned, it shows remarkable adaptation capabilities, especially on domain-shift benchmarks like Ref-GTA.

**Strengths:**

- The core contribution—a symbolic executor that blends imperative Python with differentiable, soft-logic operators (Table 2)—is highly novel and effectively tackles the problem of uncertainty propagation in multi-step reasoning.

- The framework's differentiability is a key advantage. The ability to fine-tune the system for a new domain (as shown on Ref-GTA, where neuro-symbolic fine-tuning achieved 69.9% vs. 32.6% for standard VLM fine-tuning) is a significant step forward from inference-only models.

- By operating on continuous scores rather than crisp decisions, NePTune avoids the cascading error problem that plagues many multi-step reasoning pipelines.

- NePTune demonstrates state-of-the-art zero-shot performance against comparable methods (e.g., 92.65% vs 36.05% for ViperGPT on CLEVR) and massive gains on challenging domain-shift benchmarks (Ref-GTA), proving the practical value of its design.

- The paper is extremely well-written, and its contributions are clearly articulated and well-situated against prior work.

**Weaknesses:**

- The entire system still hinges on an LLM's ability to correctly parse the natural language query into a logically sound Python program. The paper notes that errors shift from "syntax" to "logic" (Sec 5), but a logical error in the generated program is just as fatal as a syntax error. A deeper analysis of the types of logical errors the LLM still makes would be valuable.

- The score function for relational queries (num_objects=2) involves VLM queries that result in an $N \times N$ matrix, where N is the number of detected objects. This approach seems to scale quadratically with the number of objects in the scene. It's unclear how the system would perform in highly cluttered real-world scenes with hundreds of object proposals.

**Questions:**

- The ablation study (Table 10) shows a massive +19.19% improvement from adding "Imperative Reasoning" on top of declarative logic. Could the authors provide a concrete example of a query from CLEVR-Humans that requires this imperative logic (e.g., loops or complex conditionals) and which a purely declarative approach fails to solve?

- Regarding the $N \times N$ matrix for relational queries: What is the practical limit for N (number of objects) before the VLM grounding and subsequent reasoning become computationally prohibitive? Have the authors tested this on more cluttered datasets, like COCO?

- The fine-tuning results on Ref-GTA are a key strength. To clarify: what exactly is being fine-tuned? Is it only the VLM backbone? Or are the parameters of the soft-logic operators (e.g., the temperature $\tau$ and margin $\gamma$ from Table 2) also learned during the neuro-symbolic fine-tuning process?

---

> ### Author Response · Authors · 2025-11-24
> **Response to Reviewer 9EoX**
>
> We thank reviewer R-9EoX for their insightful review and for describing the core contribution as "highly novel" and the fine-tuning results as a "significant step forward."
>
> > **W1**: The entire system still hinges on an LLM's ability to correctly parse the natural language query into a logically sound Python program. The paper notes that errors shift from "syntax" to "logic" (Sec 5), but a logical error in the generated program is just as fatal as a syntax error. A deeper analysis of the types of logical errors the LLM still makes would be valuable.
>
> Thanks for this suggestion. We conducted a human evaluation of the generated programs on 600 examples (200 questions each from Ref-Adv, Ref-GTA, and CLEVR-Humans; three evaluators per dataset). Across the three datasets, 87.5–97.5 percent of programs are judged fully correct, syntax errors are essentially absent, and logical errors are very rare (around 1–2 percent of programs). Please see General Response 2 for more details.
>
> > **Q1**: The ablation study (Table 10) shows a massive +19.19% improvement from adding "Imperative Reasoning" on top of declarative logic. Could the authors provide a concrete example of a query from CLEVR-Humans that requires this imperative logic (e.g., loops or complex conditionals) and which a purely declarative approach fails to solve?
>
>
> In first-order logic based declarative languages used in the compared baselines (LEFT, NeSyCoCo) where declarative logic is similar to NePTune, declarative language supports first-order logic over fixed predicates, but CLEVR-Humans queries require reasoning over concepts themselves, which need to iterate over predicates and require a second-order logic to be expressed that is not expressible in these declarative languages. An example of these types of queries is shown in Figure 4, Example 1: "How many shapes have the same material as the only brown shape?". The program must first select the unique brown object, bind its material to a variable, and then iterate over all objects to count those whose material matches it. A declarative first-order logical language, such as LEFT, can count objects with a given attribute, but cannot express counting over predicates in a straightforward way. Below, we demonstrate the program generated for each method.
>
> **NePTune**
> ```python
> is_brown = score("Is the main object in the red bounding box brown?", num_objects=1)
>
> same_material = score("Does the main object in the red bounding box have the same material\
> as the main object in the green bounding box?", num_objects=2)
>
> same_material_as_brown = (same_material("x1", "x2") & is_brown("x1").iota("x1"))
>
> shapes = set()
> for i in range(objects_count):
>     if same_material_as_brown[i]:
>         shape = query("What is the shape of the main object in the red bounding box?", object_id=i)
>         shapes.add(shape)
>
> return len(shapes)
> ```
>
> **LEFT**
> ```python
> count(Object, lambda x: same_material(x, iota(Object, lambda y: brown(y) and shape(y))) and shape(x))
> ```
>
>
> > **Q1,W2:** Regarding the matrix for relational queries: What is the practical limit for N (number of objects) before the VLM grounding and subsequent reasoning become computationally prohibitive? Have the authors tested this on more cluttered datasets, like COCO?
>
>  Based on your feedback, we provide a quantitative latency analysis in Appendix G. We contextualize the $O(N^2)$ relational cost by noting that the number of objects $N$ in RefCOCO-Adv (COCO-based) benchmarks is small (average 3.6), so this term is inexpensive in practice. For cluttered COCO-like scenes, we first restrict $N$ using text-driven object proposals, Grounding DINO with a low threshold for high recall (keywords extracted with the LLM during program generation), so NePTune only reasons over a focused set of candidate objects rather than all detections in the image. Our new heatmaps in Appendix G show that high latency only appears for **rare** cases with very large N; for N under 10, queries are consistently fast. Please see our detailed answer in General Response 1.
>
> > **Q3**: The fine-tuning results on Ref-GTA are a key strength. To clarify: what exactly is being fine-tuned? Is it only the VLM backbone? Or are the parameters of the soft-logic operators (e.g., the temperature and margin from Table 2) also learned during the neuro-symbolic fine-tuning process?
>
> During fine-tuning, only the added LoRA weights on the backbone VLM are updated (LoRA hyperparameters are described in Appendix F). The soft-logic only has hyperparameters (i.e., temperature and margin are not learned).

---

### Official Review · Reviewer_Nt6P · 2025-11-02

**Soundness:** 2
**Presentation:** 3
**Contribution:** 2
**Rating:** 4
**Confidence:** 2

**Summary:**

This paper introduces NePTune, a neuro-symbolic framework for vision-language reasoning. NePTune translates natural language queries into executable Python programs that use a hybrid execution model. This model combines imperative Python control flow with differentiable, soft logic operators that reason over uncertainty scores from a Vision-Language Model (VLM). The framework operates zero-shot by using an LLM to generate programs and a VLM to ground atomic concepts. It also supports fine-tuning for domain adaptation, demonstrating strong compositional generalization, especially on domain-shift benchmarks.

**Strengths:**

A substantive assessment of the strengths of the paper, touching on each of the following dimensions: originality, quality, clarity, and significance. We encourage reviewers to be broad in their definitions of originality and significance. For example, originality may arise from a new definition or problem formulation, creative combinations of existing ideas, application to a new domain, or removing limitations from prior results.

1. On the Ref-GTA domain-shift benchmark, the VLM backbone (InternVL2.5) achieves only 6.95% accuracy, while NePTune achieves 69.69%. This demonstrates a highly robust decoupling of reasoning from perception.
2. The framework integrates imperative Python with soft, differentiable logical operators. This is more expressive than declarative-only systems and more robust than crisp-logic pipelines.
3. The soft-logic operators make the pipeline differentiable, enabling fine-tuning.

**Weaknesses:**

A substantive assessment of the weaknesses of the paper. Focus on constructive and actionable insights on how the work could improve towards its stated goals. Be specific, avoid generic remarks. For example, if you believe the contribution lacks novelty, provide references and an explanation as evidence; if you believe experiments are insufficient, explain why and exactly what is missing, etc.

1. Performance is fundamentally limited by the VLM's ability to ground atomic concepts.
2. While more robust, the LLM-based program generator is still a point of failure. The authors provide a clear example of this in Appendix A.3, where the LLM generates a program with incorrect logic and arity, causing an execution failure.
3. The framework is complex, involving multiple serial steps. This multi-component pipeline is likely to have significantly higher latency than a single VLM forward pass.

**Questions:**

Please list up and carefully describe any questions and suggestions for the authors. Think of the things where a response from the author can change your opinion, clarify a confusion or address a limitation. This is important for a productive rebuttal and discussion phase with the authors.

1. How frequent are logical program generation failures, and does NePTune have any mechanism to detect or recover from them?
2. Could the authors provide a rough estimate of the end-to-end wall-clock time for a complex query? How does this multi-step pipeline's latency compare to a single forward pass of the backbone VLM?

---

> ### Author Response · Authors · 2025-11-23
> **Response to Reviewer Nt6P**
>
> We thank our reviewer for highlighting NePTune's "highly robust decoupling of reasoning from perception," especially on the Ref-GTA domain shift benchmark. Here are happy to address your concerns:
>
> > **W1**: Performance is fundamentally limited by the VLM's ability to ground atomic concepts.
>
>  We agree that NePTune inherits the perception strengths and weaknesses of its VLM backbone, since atomic predicates are scored by the underlying model. However, the key point is that NePTune employs the VLM for simple atomic perception queries rather than complex compositional queries. Comparing the results in Tables 4 and 8 shows that VLMs are better at recognizing these simpler perception-level tasks. For instance, InternVL2.5 achieves 0.945 micro-F1 on atomic CLEVR concepts, whereas its accuracy on CLEVR-Ref as a monolithic model is only 27%. This indicates that most failures stem from compositional reasoning rather than from atomic concept grounding. Furthermore, our symbolic fine-tuning improves concept grounding and yields more accurate scores, resulting in larger gains on compositional benchmarks, suggesting that better atomic grounding (stronger VLMs) is complementary to NePTune rather than a limitation of the framework itself. In summary, our contribution on compositional generalization is orthogonal to the pretrained backbones, and we can swap in stronger, faster VLMs.
>
>
> > **W2/Q1**: How frequent are logical program generation failures, and does NePTune have any mechanism to detect or recover from them?
>
> Based on your feedback, we added a new **Appendix H** to evaluate program failures. Please refer to General Response 2 for more details. Briefly,  We added a 600-example human evaluation across Ref-Adv, Ref-GTA, and CLEVR-Humans (three evaluators per dataset) in Appendix H. Our evaluation showed that 87% (Ref-Adv) to 97% (Ref-GTA) of LLM-generated programs are judged correct, and logical errors account for fewer than 2% of programs overall. These results indicate that program generation failures are infrequent in practice.
>
> NePTune handles failures in two ways. First, we catch all syntax and runtime issues at execution time by running each generated program inside a guarded wrapper and treating any exception as a program failure. In such cases, we rely on the backbone models' direct prediction. Second, we apply pattern-based fixes for a common semantic issue, mismatched `red` versus `green` bounding boxes in `score` questions, which already corrected all color-swap cases we observed in our human evaluation samples.
>
>
>
> > **W3, Q2**: The framework is complex, involving multiple serial steps. This multi-component pipeline is likely to have significantly higher latency than a single VLM forward pass.
>
> Based on your feedback, we added a new **Appendix G** for lantecy analysis. We agree that NePTune introduces extra computation compared to a single-pass VLM.  On RefCOCO-Adv and A6000 GPU, a single InternVL2.5 1B forward pass takes on average 0.90 seconds, while a NePTune execution takes 3.63 seconds. For InternVL2.5 8B, it takes, on average, 1.48 seconds for a single VLM pass and 12.35 seconds for NePTune, so the raw cost is a few times that of a single VLM evaluation. There are some ways to improve this latency. In practice, all queries reuse a limited set of images and part of the atomic query. This allows the VLM's prefix cache to be reused. In addition, batch processing is another improvement. By making these changes, NePTune's latency drops to 1.30 seconds for the 1B backbone and 8.19 seconds for the 8B backbone. The dominant cost remains the repeated VLM calls for atomic predicates, so the overall latency is a small multiple of a single VLM pass. We will make this trade-off between accuracy and latency explicit in the paper, noting that the moderate overhead comes with substantial gains in compositional robustness compared to the same backbone. Also, we can use smaller models and gain competitive results with larger models, as shown in Tables 6 and 7.

---

### Author Response · Authors · 2025-11-23
**General Response (2/2)**

> **2. On LLM Program Generator Failures (R-Nt6P, R-9EoX):**

Our revised manuscript now includes **Appendix H**, which includes human evaluation results on the correctness of Neptune’s programs.

**In summary**, 97% of programs generated for CLEVR-Humans were correct, compared with 87% for Ref-Adv programs.

There are two distinct types of issues:
(1) syntactic or runtime failures of the generated program. For the first error type, targeting standard Python with simple grounding functions (`score`, `query`) significantly reduces brittleness compared to systems that require specialized languages (e.g., NAVER, NeSyCoCo). As shown in the Choice of Symbolic Program subsection, NePTune achieves higher execution success rates (fewer runtime errors) than other similar approaches. We believe this is because generating Python code is easier for current LLMs than generating domain-specific languages. Among Python-based systems, NePTune is also more robust than ViperGPT-style pipelines. ViperGPT often builds and indexes dynamic lists of objects, leading to runtime filtering and indexing failures. In contrast, NePTune reasons globally over a fixed set of bounding boxes with soft logic operators on tensors, so the program structure does not rely on fragile positional indexing.

(2) semantic errors where the program is executable but logically wrong. To shed light on this second type, we ran a 600-example human study (200 questions each on Ref-Adv, Ref-GTA, and CLEVR-Humans; three evaluators; see Appendix H).
Evaluators labeled each generated program as *correct*, *syntax error*, *wrong atomic question*, *wrong logic*, or *other*. The aggregate results are:

| Dataset      | Correct | Syntax | Atomic | Logic | Other | Acc. |
| ------------ | ------- | ------ | ------ | ----- | ----- | --------- |
| Ref-Adv      | 175     | 0      | 21     | 3     | 1     | 87.50%    |
| Ref-GTA      | 187     | 0      | 12     | 1     | 1     | 93.50%    |
| CLEVR-Humans | 195     | 0      | 0      | 5     | 0     | 97.50%    |

Overall, 87% to 97% of programs are judged fully correct, syntax errors are essentially absent, and the remaining failures are a few semantic issues:
- On **Ref-Adv** and **Ref-GTA**, the main source of errors is *wrong atomic questions* (e.g., dropping plural/group cues, missing attributes, or slightly mis-phrased `score` prompts while the overall structure is right). In Ref-Adv and Ref-GTA, there were also two instances of mismatched bounding box color, which our pattern-matching method captured and fixed.
- On **CLEVR-Humans**, as we expect for complex human queries, errors are mostly logical. They mostly involve misuse of the iota normalization for indefinite descriptions or miscounting when the reference object should be excluded from a set.

---

### Author Response · Authors · 2025-11-23
**General Response (1/2)**

We sincerely thank all reviewers for their time and constructive feedback. We are delighted that reviewers found our core contributions to be strong:

- The hybrid (Python + soft-logic) execution model is "**highly novel**" (R-9EoX).
- The differentiability for fine-tuning is a "**key advantage**" (R-9EoX) and "**appealing**" (R-fRHk).
- The results on the Ref-GTA domain-shift benchmark are "**massive**" (R-9EoX) and demonstrate a "**highly robust decoupling**" (R-Nt6P).

Based on the feedback, we have identified two common themes, which we address here. We provide point-by-point replies to all other questions in the individual reviewer responses.

> **1. On computational cost, latency, and O($N^2$) scaling (R-Nt6P, R-9EoX, R-fRHk).**

We thank all reviewers for their interest in the latency of NePTune. In response, we added a detailed latency analysis in **Appendix G**, which we summarize here.

**High–level averages.** We analyzed the average latency of NePTune compared to the backbone models. On Ref-Adv and A6000 GPU, a single InternVL2.5 1B forward pass takes 0.90 seconds, while a NePTune execution takes 3.63 seconds. For InternVL2.5 8B, it takes 1.48 seconds for a single VLM pass and 12.35 seconds for NePTune, so the raw cost is a few times a single VLM evaluation. There are some ways to improve this latency. In practice, all queries reuse the same image and part of the atomic query. This allows the VLM's prefix cache to be reused. In addition, batch processing is another improvement. By making these changes, NePTune's latency drops to 1.30 seconds for the 1B backbone **(81% answered under 2s)** and 8.19 seconds for the 8B backbone (57% answered under 5s). The dominant cost remains the repeated VLM calls for atomic predicates, so the overall latency is a small multiple of a single VLM pass. We will make this trade-off between accuracy and latency explicit in the paper, noting that the moderate overhead comes with substantial gains in compositional robustness compared to the same backbone.


**Scaling and typical N.** Latency depends on the number of proposal objects ($N$) and the number of concepts ($K$), with relational reasoning scaling as O($N^2$) in $N$. In practice, $N$ is generally small. We select candidates using Grounding DINO with a low threshold and keywords extracted by the program generator, so $N$ reflects only semantically relevant objects rather than all detections in a cluttered scene. On Ref-Adv, the **average is $N \approx 3.6$** (median 3, maximum 18). About 94% of queries have ($N \le 6$) and only about (1%) have ($N \ge 10$), so very large ($N$) cases are rare.  Heatmaps in Figure 10 confirm this trend. The highest latency values appear at large $N$, not merely large $K$. In one extreme bin, the 8B model at ($N$ = 14, $K$ = 5) reaches about 176 s, while the corresponding 1B case is about 12 s.


**Mitigations and trade-off.** When latency is critical for a given target budget ($T$), we can take two additional steps. First, we can estimate runtime from the number of relational `score` calls and the value of $N$, then dynamically choose between a direct VLM pass and NePTune execution. Second, we can use the fine–tuned 1B backbone, which achieves performance comparable to the 8B model on Ref–Adv and Ref–GTA (Tables 6 and 7) while maintaining significantly lower latency.

Overall, NePTune’s extra cost is **concentrated in a small number of cases** with unusually large $N$, where relation scoring scales as $O(N^2)$. For the **vast majority of Ref-Adv queries**, $N$ is small (**94% have $N \le 6$**), so **after caching and batching, the runtime remains a modest multiple of a single backbone pass**. When **latency is critical**, we can use a **simple runtime estimate** to choose between NePTune execution and a **faster backbone** option, or we can **prefer the fine-tuned 1B backbone** to **further reduce runtime** while maintaining **comparable accuracy**.

Update: We uploaded the updated paper with the new analysis results and improvements to address the reviewers' concerns.

---

### Meta-Review · Area_Chair_AKgs · 2026-01-06

**Summary:**

The paper presents NePTune. NePTune is a neuro-symbolic framework designed to improve the compositional reasoning of VLMs by dynamically translating natural language queries into executable Python programs. This training-free framework integrates foundational vision models with symbolic reasoning using a hybrid execution model that incorporates soft logic operators to handle VLM uncertainty. This results in significant improvements in compositional generalization across visual reasoning benchmarks. Overall, the reviewers have mixed opinions, leaning towards a weak accept. The more negative review raises concerns that the resulting system is more complex than the VLM alone and justifies this with higher running times and failures in the mapping to the Python code. While correct, this can also be raised about VLMs. While the positive reviews also point out that the scaling is quadratic in the number of objects in the scene, they highlight the strong performance of the NePTune. I do agree on this. However, I think there are other related work (whether using Pyhton or not, this is baout compiling to a determinsitic program) that should be mentioned., such as Wüst et al: Pix2Code: Learning to Compose Neural Visual Concepts as Programs. UAI 2024: 3829-3852 and Chen et al: ComPhy: Compositional Physical Reasoning of Objects and Events from Videos. ICLR 2022

**Reviewer Concerns:**

The concerns about system complexity and higher running times raised by the negative reviewer were not really addressed but are also rather natural points of both, VLMs and neuro-symbolic approaches. The combinatorial complexity is also not solved by VLMs but indeed are also inherent trade-offs of the approach presented. The quadratic scaling issue with respect to the number of objects in the scene, noted by the positive reviewers, is also an outstanding limitation that was acknowledged but not fully addressed. However, the strong performance results presented by the authors appear to have satisfied the positive reviewers regarding the practical benefits of the approach. The main outstanding concern is the incomplete discussion of related work on compiling to deterministic programs. The authors should include more comprehensive coverage of related approaches, particularly work on learning to compose neural concepts as programs and compositional physical reasoning, to properly situate their contributions within the existing research landscape.

**Reviewer Scores:**

I believe the positive reviewers would have raised their scores even more, as indicated in their reviews. The strong performance results and the authors' responses appear to have strengthened their positive views. The more negative reviewer would likely maintain the negative position, as the core concerns about system complexity, running times, and mapping failures are inherent characteristics of the approach presented.

---

### Decision · Program_Chairs · 2026-01-26

Accept (Poster)